# VRA: Variational Rectified Activation for Out-of-distribution Detection

**Mingyu Xu**[1,2]**, Zheng Lian**[1]*, **Bin Liu**[1,2]**, Jianhua Tao**[3,4]

[1]The State Key Laboratory of Multimodal Artificial Intelligence Systems,
Institute of Automation, Chinese Academy of Sciences
[2]School of Artificial Intelligence, University of Chinese Academy of Sciences
[3]Department of Automation, Tsinghua University
[4]Beijing National Research Center for Information Science and Technology, Tsinghua University
{xumingyu2021, lianzheng2016}@ia.ac.cn

## Abstract

Out-of-distribution (OOD) detection is critical to building reliable machine learning systems in the open world. Researchers have proposed various strategies to reduce model overconfidence on OOD data. Among them, ReAct is a typical and effective technique to deal with model overconfidence, which truncates high activations to increase the gap between in-distribution and OOD. Despite its promising results, is this technique the best choice? To answer this question, we leverage the variational method to find the optimal operation and verify the necessity of suppressing abnormally low and high activations and amplifying intermediate activations in OOD detection, rather than focusing only on high activations like ReAct. This motivates us to propose a novel technique called "Variational Rectified Activation (VRA)", which simulates these suppression and amplification operations using piecewise functions. Experimental results on multiple benchmark datasets demonstrate that our method outperforms existing post-hoc strategies. Meanwhile, VRA is compatible with different scoring functions and network architectures. Our code is available at https://github.com/zeroQiaoba/VRA.

## 1 Introduction

Systems deployed in the real world often encounter out-of-distribution (OOD) data, i.e., samples from an irrelevant distribution whose label set has no interaction with the training data. Most of the existing systems tend to generate overconfident estimations for OOD data, seriously affecting their reliability [1]. Therefore, researchers propose the OOD detection task, which aims to determine whether a sample comes from in-distribution (ID) or OOD. This task allows the model to reject recognition when faced with unfamiliar samples. Considering its importance, OOD detection has attracted increasing attention from researchers and has been applied to many fields with high-safety requirements such as autonomous driving [2] and medical diagnosis [3].

In OOD detection, existing methods can be roughly divided into two categories: methods requiring training and post-hoc strategies. The first category identifies OOD data by training-time regularization [4, 5] or external OOD samples [6, 7]. But they require more computational resources and are inconvenient in practical applications. To this end, researchers propose post-hoc strategies that directly use pretrained models for OOD detection. Due to their ease of implementation, these methods have attracted increasing attention in recent years. Among them, React [8] is a typical post-hoc strategy that truncates abnormally high activations to increase the gap between ID and OOD, thereby improving detection performance. But is this operation the best choice for widening the gap?

---

*Corresponding Author

To answer this question, we use the variational method to solve for the optimal operation. Based on this operation, we reveal the necessity of suppressing abnormally low and high activations and amplifying intermediate activations in OOD detection. Then, we propose a simple yet effective strategy called "Variational Rectified Activation (VRA)", which mimics suppression and amplification operations using piecewise functions. To verify its effectiveness, we conduct experiments on multiple benchmark datasets, including CIFAR-10, CIFAR-100, and the more challenging ImageNet. Experimental results demonstrate that our method outperforms existing post-hoc strategies, setting new state-of-the-art records. The main contributions of this paper can be summarized as follows:

- (**Theory**) From the perspective of the variational method, we find the best operation for maximizing the gap between ID and OOD. This operation verifies the necessity of suppressing abnormally low and high activations and amplifying intermediate activations.

- (**Method**) We propose a simple yet effective post-hoc strategy called VRA. Our method is compatible with various scoring functions and network architectures.

- (**Performance**) Experimental results on benchmark datasets demonstrate the effectiveness of our method. VRA is superior to existing post-hoc strategies in OOD detection.

## 2 Methodology

### 2.1 Problem Definition

Let $\mathcal{X}$ be the input space and $\mathcal{Y}$ be the label space with $c$ distinct categories. Consider a supervised classification task on a dataset containing $N$ labeled samples $\{\mathbf{x}, y\}$, where $y \in \mathcal{Y}$ is the ground-truth label for the sample $\mathbf{x} \in \mathcal{X}$. Ideally, all test samples come from the same distribution as the training data. But in practice, the test sample may come from an unknown distribution, such as an irrelevant distribution whose label set has no intersection with $\mathcal{Y}$. In this paper, we use $p_{\text{in}}$ to represent the marginal distribution of $\mathcal{X}$ and $p_{\text{out}}$ to represent the distribution of OOD data. In this paper, we aim to determine whether a sample comes from ID or OOD.

### 2.2 Motivation

Among all methods, ReAct is a typical and effective post-hoc strategy [8]. Suppose $h(\mathbf{x}) = \{z_i\}_{i=1}^m$ is the feature vector of the penultimate layer and $m$ denotes the feature dimension. For convenience, we use $z$ as shorthand for $z_i$. ReAct truncates activations above a threshold $c$ for each $z$:

$$g(z) = \min(z, c), \tag{1}$$

where $c = \infty$ is equivalent to the model without truncation. ReAct has demonstrated that this truncation operation can increase the gap between ID and OOD [8]:

$$\mathbb{E}_{\text{in}}[g(z)] - \mathbb{E}_{\text{out}}[g(z)] \geq \mathbb{E}_{\text{in}}[z] - \mathbb{E}_{\text{out}}[z]. \tag{2}$$

Despite its promising results, is this strategy the best option for widening the gap between ID and OOD? In this paper, we attempt to answer this question with the help of the variational method.

### 2.3 VRA Framework

To find the best operation, we should optimize the following objectives:

- Maximize the gap between ID and OOD.
- Minimize the modification brought by the operation to maximally preserve the input.

The final objective function is calculated as follows:

$$\min_g \mathcal{L}(g) = \mathbb{E}_{\text{in}}[(g(z) - z)^2] - 2\lambda \left( \mathbb{E}_{\text{in}}[g(z)] - \mathbb{E}_{\text{out}}[g(z)] \right), \tag{3}$$

where $\lambda$ controls the trade-off between two losses. To solve for Eq. 3, we first make a mild assumption to ensure the function space $\mathcal{G}$ is sufficiently complex.

**Assumption 1** *We assume* $\mathbb{E}_{in}[|z|]$, $\mathbb{E}_{out}[|z|]$, $\mathbb{E}_{in}[z^2]$, *and* $\mathbb{E}_{out}[z^2]$ *exist. Let* $\mathcal{G}$ *be a Hilbert space:*

$$\mathcal{G} = \{g(z) | \mathbb{E}_{in}[|g(z)|], \mathbb{E}_{out}[|g(z)|], \mathbb{E}_{in}[g(z)^2], \mathbb{E}_{out}[g(z)^2] < \infty\}. \tag{4}$$

*This space is sufficiently complex containing most functions, such as identity functions, constant functions, and all bounded continuous functions. Then, we define the inner product of* $\mathcal{G}$ *as follows:*

$$< g_a(z), g_b(z) >= \int g_a(z) g_b(z) p_{in}(z) dz. \tag{5}$$

Combining this assumption, the equivalent equation of Eq. 3 is:

$$\min_{g \in \mathcal{G}} \mathcal{L}(g) = \int (g(z) - z)^2 \, p_{\text{in}}(z) - 2\lambda g(z)(p_{\text{in}}(z) - p_{\text{out}}(z)) dz. \tag{6}$$

Then, we leverage the variational method to solve for the functional extreme value. We mark $g^*(\cdot)$ as the optimal solution. $\forall f(\cdot) \in \mathcal{G}$ and $\forall \epsilon \in \mathbb{R}$, we then have:

$$\mathcal{L}(g^*) \leq \mathcal{L}(g^* + \epsilon f). \tag{7}$$

It can be converted to:

$$\int (g^*(z) - z)^2 \, p_{\text{in}}(z) - 2\lambda g^*(z)(p_{\text{in}}(z) - p_{\text{out}}(z)) dz \tag{8}$$

$$\leq \int (g^*(z) + \epsilon f(z) - z)^2 \, p_{\text{in}}(z) - 2\lambda(g^*(z) + \epsilon f(z))(p_{\text{in}}(z) - p_{\text{out}}(z)) dz. \tag{9}$$

Then, we have:

$$\epsilon^2 \int f^2(z) p_{\text{in}}(z) dz + 2\epsilon \int f(z) \left( g^*(z) - z - \lambda \left( 1 - \frac{p_{\text{out}}(z)}{p_{\text{in}}(z)} \right) \right) p_{\text{in}}(z) dz \geq 0. \tag{10}$$

Combining Assumption 1 and the arbitrariness of $\epsilon$, we can get:

$$\int f(z) \left( g^*(z) - z - \lambda \left( 1 - \frac{p_{\text{out}}(z)}{p_{\text{in}}(z)} \right) \right) p_{\text{in}}(z) dz = 0. \tag{11}$$

Considering Assumption 1 and the arbitrariness of $f(z)$, we have:

$$g^*(z) - z - \lambda \left( 1 - \frac{p_{\text{out}}(z)}{p_{\text{in}}(z)} \right) = 0. \tag{12}$$

Therefore, the optimal activation function is:

$$g^*(z) = z + \lambda \left( 1 - \frac{p_{\text{out}}(z)}{p_{\text{in}}(z)} \right). \tag{13}$$

To verify its effectiveness, we compare the optimal function $g^*(\cdot)$ with the unrectified function $g(z) = z$. Since $g^*(\cdot)$ is the optimal solution, it should get a smaller value in Eq. 3:

$$\mathbb{E}_{\text{in}}[(g^*(z) - z)^2] - 2\lambda \left( \mathbb{E}_{\text{in}}[g^*(z)] - \mathbb{E}_{\text{out}}[g^*(z)] \right) \leq \mathbb{E}_{\text{in}}[(z - z)^2] - 2\lambda \left( \mathbb{E}_{\text{in}}[z] - \mathbb{E}_{\text{out}}[z] \right). \tag{14}$$

The equivalent equation of Eq. 14 is:

$$\left( \mathbb{E}_{\text{in}}[g^*(z)] - \mathbb{E}_{\text{out}}[g^*(z)] \right) - \left( \mathbb{E}_{\text{in}}[z] - \mathbb{E}_{\text{out}}[z] \right) \geq \frac{1}{2\lambda} \mathbb{E}_{\text{in}}[(g^*(z) - z)^2]. \tag{15}$$

It shows that $g^*(\cdot)$ enlarges the gap between ID and OOD by at least $\frac{1}{2\lambda} \mathbb{E}_{\text{in}}[(g^*(z) - z)^2] \geq 0$.

## 2.4 Practical Implementations

Through theoretical analysis, we have found the optimal operation $g^*(\cdot)$ that can maximize the gap between ID and OOD. But in practice, this operation depends on the specific expressions of $p_{\text{in}}$ and $p_{\text{out}}$. Estimating these expressions is a challenging task given that OOD data comes from unknown distributions [9]. This drives us to look for more practical implementations.

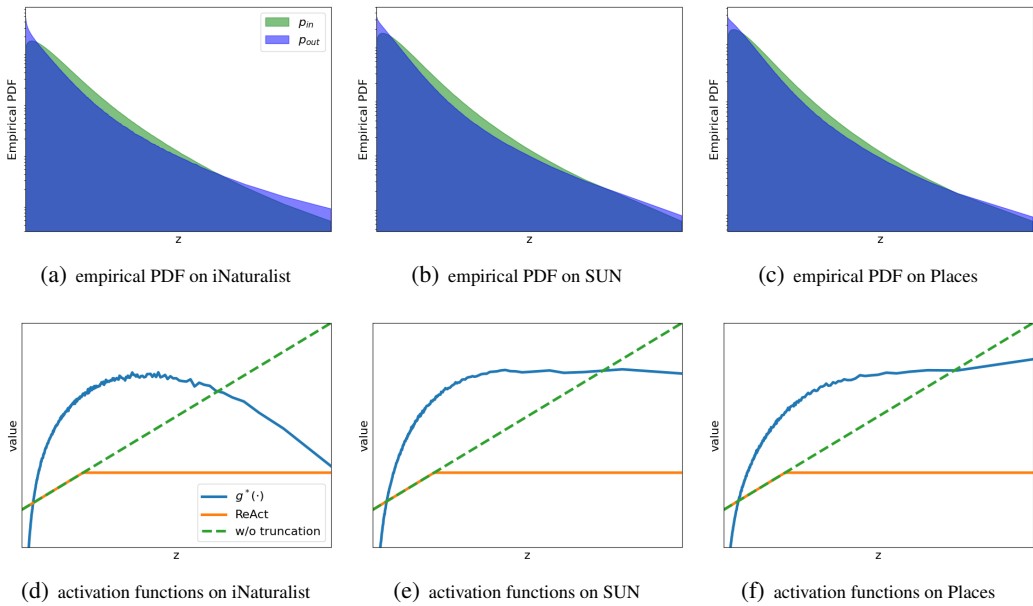

(a) empirical PDF on iNaturalist      (b) empirical PDF on SUN      (c) empirical PDF on Places

(d) activation functions on iNaturalist    (e) activation functions on SUN    (f) activation functions on Places

Figure 1: Empirical PDFs for $p_{\text{in}}(\cdot)$ and $p_{\text{out}}(\cdot)$, and visualization of different activation functions. We treat ImageNet as ID data and select multiple OOD datasets for visualization.

For this purpose, we treat ImageNet as ID data and select multiple OOD datasets. We first use histograms to approximate the probability density functions of $p_{\text{in}}$ and $p_{\text{out}}$. Then, we compute $g^*(\cdot)$ and compare it with ReAct, whose threshold is set to the $90^{th}$ percentile of activations estimated on ID data, consistent with the original paper [8]. Experimental results are shown in Figure 1. Compared with the model without truncation, we observe that ReAct suppresses high activations (see Figure 1(d)~1(f)). Unlike ReAct, the optimal operation $g^*(\cdot)$ further demonstrates the necessity of suppressing abnormally low activations in OOD detection. To mimic such operations, we design a piecewise function called VRA:

$$\text{VRA}(z) = \begin{cases} 0, z < \alpha \\ z, \alpha \le z \le \beta \\ \beta, z > \beta \end{cases} ,$$

where $\alpha$ and $\beta$ are two thresholds for determining low and high activations. Obviously, $\alpha = 0$ and $\beta = \infty$ represent models without activation truncation; $\alpha = 0$ and $\beta > 0$ represent models equivalent to ReAct. Therefore, our method is a more general operation. Since different features have distinct distributions, we propose an adaptively adjusted strategy to determine $\alpha$ and $\beta$. Specifically, we predefine $\eta_\alpha$ and $\eta_\beta$ satisfying $\eta_\alpha < \eta_\beta$. Then, we treat the $\eta_\alpha$-quantile (or $\eta_\beta$-quantile) of activations estimated on ID data as $\alpha$ (or $\beta$). Meanwhile, we observe that $g^*(\cdot)$ amplifies intermediate activations in Figure 1(d)~1(f). Therefore, we propose another variant of VRA called VRA+, which further introduces a hyper-parameter $\gamma$ to control the degree of amplification:

$$\text{VRA+}(z) = \begin{cases} 0, z < \alpha \\ z + \gamma, \alpha \le z \le \beta \\ \beta, z > \beta \end{cases} .$$

## 3 Experiments

### 3.1 Experimental Setup

**Corpus Description** In line with previous works, we consider different OOD datasets for distinct ID datasets [8, 10]. For CIFAR benchmarks [11] as ID data, we use the official train/test splits for ID data and select six datasets as OOD data: Textures [12], SVHN [13], Places365 [14], LSUN-Crop [15], LSUN-Resize [15], and iSUN [16]; for ImageNet [17] as ID data, it is more challenging than

Table 1: **Main results on CIFAR benchmarks**. In this table, we compare detection performance with competitive post-hoc strategies. All methods are pretrained on ID data. We report the results for each dataset, as well as the average results across all datasets. "FR." and "AU." are abbreviations of FPR95 and AUROC. Top3 results are marked in red, and darker colors indicate better performance.

| Method | SVHN | | LSUN-C | | LSUN-R | | iSUN | | Textures | | Places365 | | Average | |
|---|---|---|---|---|---|---|---|---|---|---|---|---|---|---|
| | FR. ↓ | AU. ↑ | FR. ↓ | AU. ↑ | FR. ↓ | AU. ↑ | FR. ↓ | AU. ↑ | FR. ↓ | AU. ↑ | FR. ↓ | AU. ↑ | FR. ↓ | AU. ↑ |
| ID Dataset: CIFAR-10; Backbone: DenseNet-101 [28] | | | | | | | | | | | | | | |
| MSP [20] | 47.27 | 93.48 | 33.57 | 95.54 | 42.10 | 94.51 | 42.31 | 94.52 | 64.15 | 88.15 | 63.02 | 88.57 | 48.74 | 92.46 |
| ODIN [21] | 25.29 | 94.57 | 04.70 | 98.86 | 03.09 | 99.02 | 03.98 | 98.90 | 57.50 | 82.38 | 52.85 | 88.55 | 24.57 | 93.71 |
| Mahalanobis [22] | 06.42 | 98.31 | 56.55 | 86.96 | 09.14 | 97.09 | 09.78 | 97.25 | 21.51 | 92.15 | 85.14 | 63.15 | 31.42 | 89.15 |
| Energy [23] | 40.61 | 93.99 | 03.81 | 99.15 | 09.28 | 98.12 | 10.07 | 98.07 | 56.12 | 86.43 | 39.40 | 91.64 | 26.55 | 94.57 |
| ReAct [8] | 41.64 | 93.87 | 05.96 | 98.84 | 11.46 | 97.87 | 12.72 | 97.72 | 43.58 | 92.47 | 43.31 | 91.03 | 26.45 | 95.30 |
| KNN [24] | 13.51 | 96.68 | 30.95 | 93.82 | 11.37 | 97.72 | 10.79 | 97.91 | 24.50 | 95.19 | 63.88 | 85.00 | 25.83 | 94.39 |
| DICE [10] | 25.99 | 95.90 | 00.26 | 99.92 | 03.91 | 99.20 | 04.36 | 99.14 | 41.90 | 88.18 | 48.59 | 89.13 | 20.84 | 95.25 |
| SHE [25] | 28.12 | 94.72 | 00.76 | 99.84 | 09.73 | 98.15 | 10.99 | 97.95 | 51.98 | 83.07 | 59.35 | 84.16 | 26.82 | 92.98 |
| ASH[27] | 30.14 | 95.29 | 2.82 | 99.34 | 7.97 | 98.33 | 8.46 | 98.29 | 50.85 | 88.29 | 40.46 | 91.76 | 23.45 | 95.22 |
| **VRA** | 18.75 | 96.68 | 01.32 | 99.63 | 05.80 | 98.69 | 05.70 | 98.69 | 34.89 | 93.42 | 39.98 | 91.69 | 17.74 | 96.47 |
| **VRA+** | 13.54 | 97.45 | 02.03 | 99.56 | 06.37 | 98.72 | 06.15 | 98.71 | 27.07 | 95.03 | 39.97 | 91.96 | 15.85 | 96.91 |
| ID Dataset: CIFAR-100; Backbone: DenseNet-101 [28] | | | | | | | | | | | | | | |
| MSP [20] | 81.70 | 75.40 | 60.49 | 85.60 | 85.24 | 69.18 | 85.99 | 70.17 | 84.79 | 71.48 | 82.55 | 74.31 | 80.13 | 74.36 |
| ODIN [21] | 41.35 | 92.65 | 10.54 | 97.93 | 65.22 | 84.22 | 67.05 | 83.84 | 82.34 | 71.48 | 82.32 | 76.84 | 58.14 | 84.49 |
| Mahalanobis [22] | 22.44 | 95.67 | 68.90 | 86.30 | 23.07 | 94.20 | 31.38 | 89.28 | 62.39 | 79.39 | 92.66 | 61.39 | 50.14 | 84.37 |
| Energy [23] | 87.46 | 81.85 | 14.72 | 97.43 | 70.65 | 80.14 | 74.54 | 78.95 | 84.15 | 71.03 | 79.20 | 77.72 | 68.45 | 81.19 |
| ReAct [8] | 83.81 | 81.41 | 25.55 | 94.92 | 60.08 | 87.88 | 65.27 | 86.55 | 77.78 | 78.95 | 82.65 | 74.04 | 65.86 | 83.96 |
| KNN [24] | 23.96 | 93.99 | 70.98 | 73.37 | 76.34 | 76.69 | 70.88 | 78.58 | 37.75 | 87.48 | 95.20 | 59.70 | 62.52 | 78.30 |
| DICE [10] | 54.65 | 88.84 | 00.93 | 99.74 | 49.40 | 91.04 | 48.72 | 90.08 | 65.04 | 76.42 | 79.58 | 77.26 | 49.72 | 87.23 |
| SHE [25] | 41.89 | 90.61 | 01.06 | 99.68 | 78.18 | 73.97 | 72.73 | 76.14 | 61.49 | 76.57 | 85.33 | 70.53 | 56.78 | 81.25 |
| ASH[27] | 81.86 | 83.86 | 11.60 | 97.89 | 67.56 | 81.67 | 70.90 | 80.81 | 78.24 | 74.09 | 77.03 | 77.94 | 64.53 | 82.71 |
| **VRA** | 70.91 | 87.46 | 10.73 | 98.04 | 38.52 | 93.49 | 38.53 | 93.42 | 47.64 | 90.17 | 76.39 | 78.66 | 47.12 | 90.21 |
| **VRA+** | 62.64 | 88.70 | 19.82 | 96.33 | 28.44 | 95.47 | 28.72 | 95.18 | 40.62 | 91.57 | 79.78 | 76.42 | 43.34 | 90.61 |

CIFAR benchmarks due to larger label space and higher resolution images. To ensure non-overlapped categories between ID and OOD, we select a subset from four datasets as OOD data, in line with previous works [8, 10]: iNaturalist [18], SUN [19], Places [14], and Textures [12].

**Baselines** To verify the effectiveness of our method, we implement the following state-of-the-art post-hoc strategies as baselines: 1) MSP [20] is the most basic method that directly leverages the maximum softmax probability to identify OOD data; 2) ODIN [21] uses temperature scaling and input perturbation to increase the gap between ID and OOD; 3) Mahalanobis [22] calculates the distance from the nearest class center as the indicator; 4) Energy [23] replaces the maximum softmax probability with the theoretically guaranteed energy score; 5) ReAct [8] applies activation truncation to remove abnormally high activations; 6) KNN [24] exploits non-parametric nearest-neighbor distance for OOD detection; 7) DICE [10] leverages sparsification to select the most salient weights; 8) SHE [25] uses the energy function defined in the modern Hopfield network [26]. 9) ASH [27] removed a large portion of a sample's activation at a late layer.

**Implementation Details** Our method contains three user-specific parameters: the thresholds $\eta_\alpha$ and $\eta_\beta$, and the degree of amplification $\gamma$. We select $\eta_\alpha$ from $\{0.5, 0.6, 0.65, 0.7\}$, $\eta_\beta$ from $\{0.8, 0.85, 0.9, 0.95, 0.99\}$, and $\gamma$ from $\{0.2, 0.3, 0.4, 0.5, 0.6, 0.7\}$. Consistent with previous works [8], we use Gaussian noise images as the validation set for hyperparameter tuning. By default, we use DenseNet-101 [28] for CIFAR and ResNet-50 [29] for ImageNet. All experiments are implemented with PyTorch [30] and carried out with NVIDIA Tesla V100 GPU. To compare the performance of different methods, we exploit two widely used OOD detection metrics: FPR95 and AUROC. Among them, FPR95 measures the false positive rate of OOD data when the true positive rate of ID data is 95%; AUROC measures the area under the receiver operating characteristic curve.

## 3.2 Experimental Results and Discussion

**Main Results** To verify the effectiveness of our method, we compare VRA-based methods with competitive post-hoc strategies. Experimental results are shown in Table 1 and Table 2. We observe that our method generally achieves Top3 performance on different datasets and performs the best overall. Different from these baselines, we attempt to maximize the gap between ID and OOD by suppressing abnormally low and high activations and amplifying intermediate activations. These

Table 2: **Main results on ImageNet**. All methods are pretrained on ImageNet.

| Method | iNaturalist | | SUN | | Places | | Textures | | Average | |
|---|---|---|---|---|---|---|---|---|---|---|
| | FR. ↓ | AU. ↑ | FR. ↓ | AU. ↑ | FR. ↓ | AU. ↑ | FR. ↓ | AU. ↑ | FR. ↓ | AU. ↑ |
| Backbone: ResNet-50 [29] | | | | | | | | | | |
| MSP [20] | 54.99 | 87.74 | 70.83 | 80.86 | 73.99 | 79.76 | 68.00 | 79.61 | 66.95 | 81.99 |
| ODIN [21] | 47.66 | 89.66 | 60.15 | 84.59 | 67.89 | 81.78 | 50.23 | 85.62 | 56.48 | 85.41 |
| Mahalanobis [22] | 97.00 | 52.65 | 98.50 | 42.41 | 98.40 | 41.79 | 55.80 | 85.01 | 87.43 | 55.47 |
| Energy [23] | 55.72 | 89.95 | 59.26 | 85.89 | 64.92 | 82.86 | 53.72 | 85.99 | 58.40 | 86.17 |
| ReAct [8] | 20.38 | 96.22 | 24.20 | 94.20 | 33.85 | 91.58 | 47.30 | 89.80 | 31.43 | 92.95 |
| KNN [24] | 59.08 | 86.20 | 69.53 | 80.10 | 77.09 | 74.87 | 11.56 | 97.18 | 54.32 | 84.59 |
| DICE [10] | 25.63 | 94.49 | 35.15 | 90.83 | 46.49 | 87.48 | 31.72 | 90.30 | 34.75 | 90.78 |
| SHE [25] | 34.22 | 90.18 | 54.19 | 84.69 | 45.35 | 90.15 | 45.09 | 87.93 | 44.71 | 88.24 |
| ASH[27] | 44.57 | 92.51 | 52.88 | 88.35 | 61.79 | 85.58 | 42.06 | 89.70 | 50.32 | 89.04 |
| **VRA** | 15.70 | 97.12 | 26.94 | 94.25 | 37.85 | 91.27 | 21.47 | 95.62 | 25.49 | 94.57 |
| **VRA+** | 15.48 | 97.08 | 23.50 | 94.91 | 34.62 | 91.79 | 19.66 | 96.08 | 23.31 | 94.97 |
| Backbone: ResNetv2-101 [29] | | | | | | | | | | |
| MSP [20] | 63.69 | 87.59 | 79.98 | 78.34 | 81.44 | 76.76 | 82.73 | 75.45 | 76.96 | 79.54 |
| ODIN [21] | 62.69 | 89.36 | 71.67 | 83.92 | 76.27 | 80.67 | 81.31 | 76.30 | 72.99 | 82.56 |
| Mahalanobis [22] | 96.34 | 46.33 | 88.43 | 65.20 | 89.75 | 64.46 | 52.23 | 72.10 | 81.69 | 62.02 |
| Energy [23] | 64.91 | 88.48 | 65.33 | 85.32 | 73.02 | 81.37 | 80.87 | 75.79 | 71.03 | 82.74 |
| ReAct [8] | 49.97 | 89.80 | 65.30 | 87.40 | 73.12 | 85.34 | 80.82 | 70.53 | 67.30 | 83.27 |
| MOS [4] | 09.28 | 98.15 | 40.63 | 92.01 | 49.54 | 89.06 | 60.43 | 81.23 | 39.97 | 90.11 |
| **VRA** | 27.26 | 95.68 | 34.53 | 93.27 | 47.31 | 90.19 | 30.69 | 94.22 | 34.95 | 93.34 |
| **VRA+** | 20.81 | 97.70 | 32.89 | 92.68 | 45.83 | 90.01 | 23.88 | 95.43 | 30.85 | 93.71 |

Table 3: **Compatibility with different scoring functions**. For each ID dataset, we report the average results of its OOD datasets. We use DenseNet-101 [28] for CIFAR and ResNet-50 [29] for ImageNet.

| Method | CIFAR-10 | | CIFAR-100 | | ImageNet | | Average | |
|---|---|---|---|---|---|---|---|---|
| | FPR95 ↓ | AUROC ↑ | FPR95 ↓ | AUROC ↑ | FPR95 ↓ | AUROC ↑ | FPR95 ↓ | AUROC ↑ |
| MSP [20] | 48.73 | 92.46 | 80.13 | 74.36 | 66.95 | 81.99 | 65.27 | 82.94 |
| MSP + ReAct | 48.00 | 92.77 | 77.69 | 76.22 | 55.63 | 87.85 | 60.44 | 85.61 |
| MSP + DICE | 43.72 | 92.92 | **76.86** | **76.39** | 67.41 | 82.24 | 62.66 | 83.85 |
| MSP + VRA | **42.31** | **93.50** | 79.69 | 75.94 | **47.09** | **89.62** | **56.36** | **86.35** |
| Energy [23] | 26.55 | 94.57 | 68.45 | 81.19 | 58.41 | 86.17 | 51.14 | 87.31 |
| Energy + ReAct | 26.45 | 94.67 | 62.27 | 84.47 | 31.43 | 92.95 | 40.05 | 90.70 |
| Energy + DICE | 20.83 | 95.24 | **49.72** | 87.23 | 34.75 | 90.77 | 35.10 | 91.08 |
| Energy + VRA | **17.74** | **96.47** | 53.24 | **88.74** | **25.49** | **94.57** | **32.16** | **93.26** |
| ODIN [21] | 24.57 | 93.71 | 58.14 | 84.49 | 56.48 | 85.41 | 46.40 | 87.87 |
| ODIN + ReAct | 21.00 | 95.98 | 54.17 | 88.62 | 42.21 | 91.28 | 39.13 | 91.96 |
| ODIN + DICE | 26.05 | 94.62 | 61.39 | 83.83 | 62.89 | 84.48 | 50.11 | 87.64 |
| ODIN + VRA | **17.38** | **96.52** | **47.12** | **90.21** | **32.75** | **93.39** | **32.42** | **93.37** |

results demonstrate the effectiveness of such suppression and amplification operations in OOD detection. Meanwhile, we observe that VRA+ generally outperforms VRA, suggesting that the operation closer to the theoretical optimum solution generally can achieve better performance.

We also compare with methods that require training. MOS [4] addresses OOD detection by training-time regularization. Experimental results in Table 2 show that our method outperforms MOS with the same backbone. Meanwhile, VOS [5] is a recently advanced strategy that synthesizes virtual outliers to regularize decision boundaries during training. According to their original paper, it achieves 95.33% in AUROC and 22.47% in FPR95 on CIFAR-10. Our method outperforms VOS under the same ID data, OOD data, and network architecture (see Table 1). Therefore, VRA-based methods do not require an expensive training process but can achieve better performance in OOD detection.

**Compatibility with Scoring Functions**    In Table 3, we investigate the compatibility of VRA-based methods with different scoring functions: MSP, Energy, and ODIN. Experimental results demonstrate that our method brings performance improvements for all scoring functions and generally achieves better performance than competitive post-hoc strategies. These results verify the compatibility and effectiveness of our method in OOD detection.

Table 4: **Performance upper bound analysis**. For each ID dataset, we report the average results over multiple OOD datasets. We use DenseNet-101 [28] for CIFAR and ResNet-50 [29] for ImageNet.

| ID | Energy [23] | | VRA | | VRA-Fake-True | | VRA-True | |
|---|---|---|---|---|---|---|---|---|
| | FPR95 $\downarrow$ | AUROC $\uparrow$ | FPR95 $\downarrow$ | AUROC $\uparrow$ | FPR95 $\downarrow$ | AUROC $\uparrow$ | FPR95 $\downarrow$ | AUROC $\uparrow$ |
| CIFAR-10 | 26.55 | 94.57 | 17.74 | 96.47 | 13.27 | 97.75 | **00.96** | **99.81** |
| CIFAR-100 | 68.45 | 81.19 | 47.12 | 90.21 | 23.62 | 94.20 | **01.58** | **99.69** |
| ImageNet | 58.41 | 86.17 | 25.49 | 94.57 | 13.09 | 96.89 | **03.50** | **99.31** |

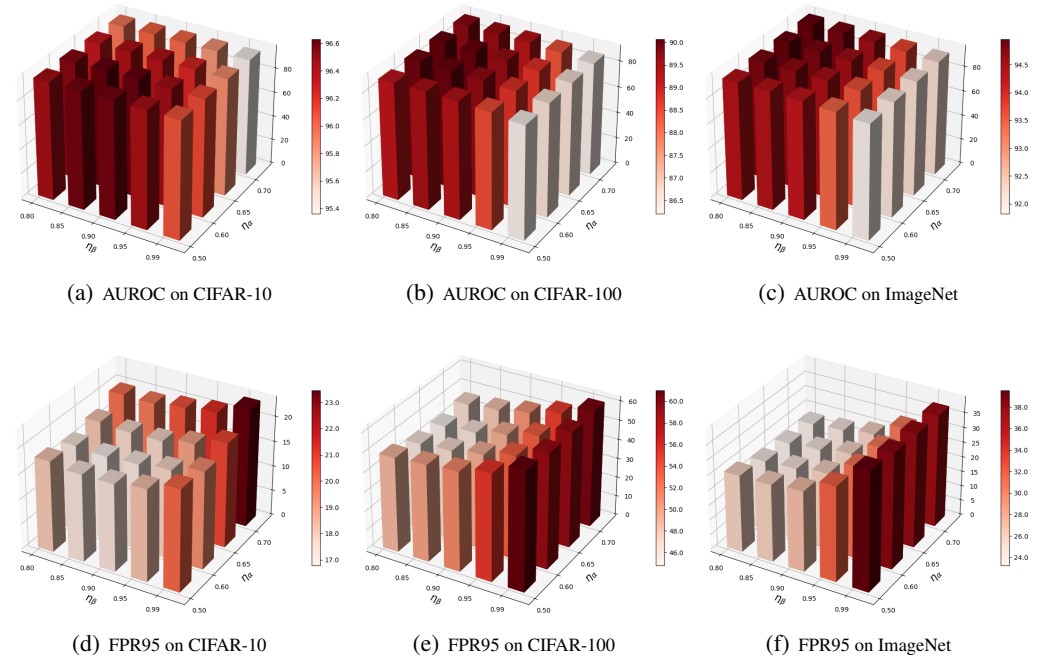

(a) AUROC on CIFAR-10    (b) AUROC on CIFAR-100    (c) AUROC on ImageNet

(d) FPR95 on CIFAR-10    (e) FPR95 on CIFAR-100    (f) FPR95 on ImageNet

Figure 2: **Parameter sensitivity analysis.** For each ID dataset, we report the average results over multiple OOD datasets. We use DenseNet-101 [28] for CIFAR and ResNet-50 [29] for ImageNet.

**Performance Upper Bound Analysis**    We propose VRA and VRA+ to approximate the optimal operation for OOD detection. But is it necessary to design other functions to get a better approximation? To answer this question, we need to reveal whether $g^*(\cdot)$ can reach the upper-bound performance. The core of estimating $g^*(\cdot)$ is to estimate the probability density functions of $p_{in}$ and $p_{out}$. To this end, we consider two ideal cases: *VRA-True* and *VRA-Fake-True*. In the first case, we assume that all ID and OOD data are known in advance; in the second case, we randomly select 1% of ID and OOD data from the entire dataset. Both cases leverage histograms to estimate $p_{in}$ and $p_{out}$ and use Eq. 13 to calculate $g^*(\cdot)$. Considering that histograms provide a piecewise form of $g^*(\cdot)$, we directly use the piecewise function to represent $g^*(\cdot)$. In Table 4, we observe that both ideal cases can achieve near-perfect results. Therefore, $g^*(\cdot)$ that increases the gap between ID and OOD can generate more discriminative features for OOD detection. In the future, we will explore other functions that can better describe the optimal operation for better performance.

**Parameter Sensitivity Analysis**    VRA uses two hyper-parameters ($\eta_\alpha$ and $\eta_\beta$) to adaptively adjust thresholds for low and high activations. In this section, we conduct parameter sensitivity analysis and reveal their impact on OOD detection. In Figure 2, we observe that our method does not perform well when $\eta_\alpha$ and $\eta_\beta$ are inappropriate. A large $\eta_\alpha$ suppresses too many low activations, while a large $\eta_\beta$ suppresses too few high activations. Therefore, it is necessary to choose proper $\eta_\alpha$ and $\eta_\beta$ for VRA.

**Role of Adaptively Adjusted Strategy**    In this paper, we adopt an adaptive strategy to automatically determine $\alpha$ and $\beta$. To verify its effectiveness, we compare this adaptive strategy with another strategy that uses fixed $\alpha$ and $\beta$ for different features. To determine these hyper-parameters, we use Gaussian

Table 5: **Role of adaptively adjusted strategy.** We use DenseNet-101 [28] for CIFAR.

| ID | Strategy | Hyper-parameters | | | | OOD Performance | |
|---|---|---|---|---|---|---|---|
| | | $\alpha$ | $\beta$ | $\eta_\alpha$ | $\eta_\beta$ | FPR95 ↓ | AUROC ↑ |
| CIFAR-10 | assign $\alpha, \beta$ | 0.50 | 1.50 | – | – | 19.44 | 96.34 |
| | assign $\eta_\alpha, \eta_\beta$ | – | – | 0.60 | 0.95 | **17.74** | **96.47** |
| CIFAR-100 | assign $\alpha, \beta$ | 0.50 | 1.50 | – | – | 56.35 | 86.09 |
| | assign $\eta_\alpha, \eta_\beta$ | – | – | 0.60 | 0.95 | **47.12** | **90.21** |

Table 6: **Compatibility with different backbones.** All methods are pretrained on ImageNet.

| Backbone | MSP | | Energy | | ReAct+Energy | | VRA+Energy | |
|---|---|---|---|---|---|---|---|---|
| | FPR95 ↓ | AUROC ↑ | FPR95 ↓ | AUROC ↑ | FPR95 ↓ | AUROC ↑ | FPR95 ↓ | AUROC ↑ |
| ResNet-18 [29] | 69.70 | 80.61 | 58.59 | 80.40 | 36.36 | 92.17 | **34.87** | **92.58** |
| ResNet-34 [29] | 68.84 | 81.19 | 57.20 | 86.84 | 32.23 | 93.08 | **30.63** | **93.46** |
| ResNet-50 [29] | 66.95 | 81.99 | 58.40 | 86.17 | 31.43 | 92.95 | **25.49** | **94.57** |
| ResNet-101 [29] | 64.70 | 82.47 | 54.84 | 87.29 | 31.68 | 93.03 | **25.80** | **94.36** |
| ResNet-152 [29] | 61.35 | 83.74 | 50.39 | 88.61 | 26.57 | 94.22 | **22.21** | **95.20** |
| VGG-16 [33] | 67.94 | 81.60 | 54.33 | 88.17 | 67.81 | 83.68 | **32.99** | **92.59** |
| VGG-16-BN [33] | 65.92 | 82.00 | 50.49 | 89.03 | 59.02 | 86.34 | **35.12** | **92.05** |
| EfficientNetV2 [34] | 57.57 | 83.96 | 75.29 | 71.10 | 48.28 | 88.01 | **43.81** | **89.76** |
| RegNet [35] | 65.37 | 82.85 | 59.46 | 85.51 | 34.65 | 92.53 | **26.18** | **94.55** |
| MobileNetV3 [36] | 67.99 | 82.14 | 60.49 | 87.80 | 60.72 | 87.82 | **56.65** | **89.30** |

noise images as the validation set, in line with previous works [8]. Experimental results in Table 5 demonstrate that our adaptive strategy outperforms this fixed strategy. The reason lies in that different features have distinct statistical distributions. Using fixed thresholds for different features will limit the performance of OOD detection.

**Compatibility with Backbones** In this section, we further verify the compatibility of our method with different backbones. For a fair comparison, all methods are pretrained on ImageNet, and we report the average results on four OOD datasets of ImageNet. Compared with competitive post-hoc strategies, experimental results in Table 6 demonstrate that our method can achieve the best performance under different network architectures. These results validate the effectiveness and compatibility of our method. Meanwhile, we observe some interesting phenomena in Table 6. ReAct [8] points out that mismatched BatchNorm [31] statistics between ID and OOD lead to model overconfidence on OOD data. In Table 6, VGG-16 and VGG-16-BN refer to models without and with BatchNorm, respectively. We observe that no matter with or without BatchNorm, ReAct cannot achieve better performance than Energy, consistent with previous findings [32]. Therefore, BatchNorm may not be the only reason for model overconfidence, and the network architecture also matters. Furthermore, Energy [23] generally outperforms MSP [20] with the exception of EfficientNetV2, which also reveals its limitation in compatibility. In the future, we will conduct an in-depth analysis to reveal the reasons behind these phenomena.

## 4 Further Analysis

Combining features with logit outputs can achieve better performance in OOD detection [37]. Therefore, we design another variant of VRA called VRA++, whose scoring function is defined as:

$$\lambda_v \sum_{i=1}^{m} g(z_i) + \log \sum_{i=1}^{c} e^{l_i}, \tag{16}$$

where $z_i, i \in [1, m]$ represents the $i$-th feature and $l_i, i \in [1, c]$ represents the $i$-th logit output. This scoring function consists of two items: (1) Since we have maximized the gap between ID and OOD $\mathbb{E}_{\text{in}}[g(z_i)] - \mathbb{E}_{\text{out}}[g(z_i)]$, we directly use the sum of all rectified features $\sum_{i=1}^{m} g(z_i)$ as the indicator; (2) We also calculate the energy score on logit outputs for OOD detection. These items are combined using a balancing factor $\lambda_v$. Unlike VRA using piecewise functions, we further test the performance of the quadratic function $g(z) = -z^2 + \alpha_v z$. By choosing a proper $\alpha_v$, this quadratic function can

Table 7: **Performance of VRA++**. All methods are based on BiT [39] and pretrained on ImageNet.

| Method | OpenImage-O | | Texture | | iNaturalist | | ImageNet-O | | Average | |
|---|---|---|---|---|---|---|---|---|---|---|
| | FR. ↓ | AU. ↑ | FR. ↓ | AU. ↑ | FR. ↓ | AU. ↑ | FR. ↓ | AU. ↑ | FR. ↓ | AU. ↑ |
| MSP [20] | 73.72 | 84.16 | 76.65 | 79.80 | 64.09 | 87.92 | 96.85 | 57.12 | 77.83 | 77.25 |
| ODIN [21] | 72.83 | 85.64 | 74.07 | 81.60 | 70.75 | 86.73 | 96.85 | 63.00 | 78.63 | 79.24 |
| Mahalanobis [22] | 64.32 | 83.10 | 14.05 | 97.33 | 64.95 | 85.70 | 70.05 | 80.37 | 53.34 | 86.63 |
| Energy [23] | 73.42 | 84.77 | 73.91 | 81.09 | 74.98 | 84.47 | 96.40 | 63.59 | 79.68 | 78.48 |
| ReAct [8] | 54.97 | 88.94 | 50.25 | 90.64 | 48.60 | 91.45 | 91.70 | 67.07 | 61.38 | 84.52 |
| ViM [37] | 43.96 | 91.54 | **04.69** | **98.92** | 55.71 | 89.30 | 61.50 | 83.87 | 41.47 | 90.91 |
| **VRA++** | **34.94** | **93.55** | 05.02 | 98.76 | **22.25** | **96.37** | **60.45** | **84.21** | **30.67** | **93.22** |

Table 8: **Comparison of VRA variants**. "Net1" and "Net2" refer to ResNet-50 and ResNetv2-101.

| Method | CIFAR-10 | | CIFAR-100 | | ImageNet (Net1) | | ImageNet (Net2) | |
|---|---|---|---|---|---|---|---|---|
| | FPR95 ↓ | AUROC ↑ | FPR95 ↓ | AUROC ↑ | FPR95 ↓ | AUROC ↑ | FPR95 ↓ | AUROC ↑ |
| VRA | 17.74 | 96.47 | 47.12 | 90.21 | 25.49 | 94.57 | 34.95 | 93.34 |
| VRA+ | 15.89 | **96.90** | 43.31 | 90.61 | 23.32 | 94.96 | 30.85 | 93.78 |
| VRA++ | **15.52** | 96.87 | **35.20** | **91.80** | **18.63** | **95.75** | **25.92** | **94.60** |

also simulate suppression and amplification operations. Finally, our scoring function is defined as:

$$-\lambda_v \sum_{i=1}^{m}(z_i^2 - \alpha_v z_i) + \log \sum_{i=1}^{c} e^{l_i}. \tag{17}$$

Among all methods, ViM [37] is a powerful strategy that combines features and logit outputs. For a fair comparison with ViM, we use the same ID data (ImageNet), OOD data (OpenImage-O [37], Texture [12], iNaturalist [18], and ImageNet-O [38]), and network architecture (BiT [39]). Experimental results in Table 7 demonstrate that VRA++ achieves better performance than ViM, verifying the scalability and high potential of our method. Meanwhile, VRA++ generally achieves the best performance among all variants (see Table 8). These results further demonstrate the necessity of combining features and logit outputs in OOD detection.

## 5   Related Work

**Post-hoc Method**   Post-hoc strategies are an important branch of OOD detection. Due to their ease of implementation, they have attracted increasing attention from researchers. Among them, MSP [20] was the most basic post-hoc strategy, which directly leveraged the maximum value of the posterior distribution as the indicator. Since then, researchers have proposed various post-hoc approaches. For example, ODIN [21] used temperature scaling and input perturbations to improve the separability of ID and OOD data. Energy [23] replaced the softmax confidence score in MSP [20] with the theoretically guaranteed energy score. Mahalanobis [22] used the minimum distance from the class centers to identify OOD data. KNN [24] was a nonparametric method that explored K-nearest neighbors. More recently, researchers have found that the reason behind model overconfidence in OOD data lies in abnormally high activations of a small number of neurons. To address this, Dice [10] used weight sparsification, while ReAct [8] exploited activation truncation. Different from these works, we further demonstrate that abnormally low activations also affect OOD detection performance. This motivates us to propose VRA to rectify the activation function.

**Activation Function**   Activation functions are an important part of neural networks [40, 41]. Previously, researchers found that neural networks with the ReLU activation function produced abnormally high activations for inputs far from the training data, harming the reliability of deployed systems [42]. To address this problem, ReAct used a truncation operation to rectify activation functions. In this paper, we propose a more powerful rectified activation function for OOD detection. Experimental results on multiple benchmark datasets demonstrate the effectiveness of our method.

**Variational Method**   The variational method is often used to solve for the functional extreme value. Its most famous application in neural networks is the variational autoencoder [43], which solves for the functional extreme value by trading off reconstruction loss and Kullback–Leibler divergence. It

has also been applied to other complex scenarios [44] and multimodal tasks [45]. In this paper, we use the variational method to find the operation that can maximize the gap between ID and OOD.

## 6   Limitation

Distributions of ID and OOD data impact the performance of VRA. In the future, we will conduct a theoretical analysis to explain the reason behind this phenomenon. Meanwhile, analogous to previous works such as ReAct and ASH, this paper mainly focuses on the pre-trained classifiers with ReLU-based activations. Although we have explored some other architectures in the appendix, future experiments in more structures are also needed.

In this paper, we treat $\max_g \mathbb{E}_{\text{in}}[g(z)] - \mathbb{E}_{\text{out}}[g(z)]$ as the core objective function derived from ReAct and $\min_g \mathbb{E}_{\text{in}}[(g(z) - z)^2]$ as the regularization term. However, there may be better regularization terms that can not only guarantee the existence of the optimal solution but also ensure that the expression of the optimal solution is easy to implement and has good interpretability. Therefore, we will explore other regularization terms for OOD detection. Meanwhile, this paper uses simple piecewise functions to approximate the complex optimal operation. In the future, we will explore other functional forms that can better describe the optimal operation.

## 7   Conclusion

This paper proposes a post-hoc OOD detection strategy called VRA. From the perspective of the variational method, we find the theoretically optimal operation for maximizing the gap between ID and OOD. This operation reveals the necessity of suppressing abnormally low and high activations and amplifying intermediate activations in OOD detection. Therefore, we propose VRA to mimic these suppression and amplification operations. Experimental results show that our method outperforms existing post-hoc strategies and is compatible with different scoring functions and network architectures. In the ideal case of knowing a small fraction of OOD samples, we can achieve near-perfect performance, demonstrating the strong potential of our method. Meanwhile, we verify the effectiveness of our adaptively adjusted strategy and reveal the impact of different hyper-parameters.

## 8   Acknowledge

This work is supported by the National Natural Science Foundation of China (No.62201572, No.61831022, No.62276259, No.U21B2010, No.62271083), Beijing Municipal Science & Technology Commission, Administrative Commission of Zhongguancun Science Park (No.Z211100004821013), Open Research Projects of Zhejiang Lab (NO. 2021KH0AB06), CCF-Baidu Open Fund (No.OF2022025).

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

## A Contrastive Learning Trained Backbones

Many post-hoc methods are conducted on classification-based backbones [8, 27]. In this section, we further perform experiments on backbones trained on contrastive learning. We use the pre-trained model from CIDER[46] and apply VRA to the normalized features, followed by KNN[24] for OOD detection. In Table 9, we observe that for the contrastive-learning-based backbone, modifying the feature values and suppressing larger and smaller activation values are also beneficial to OOD detection.

Table 9: **Comparison with contrastive learning trained backbones (SSD+[47], CSI[48], and CIDER).** We use ResNet-34 for CIFAR-100. Results of SSD+, CSI and CIDER are reported from CIDER [46].

| Method | SVHN | | Places365 | | LSUN | | iSUN | | Textures | | Average | |
|---|---|---|---|---|---|---|---|---|---|---|---|---|
| | FR. ↓ | AU. ↑ | FR. ↓ | AU. ↑ | FR. ↓ | AU. ↑ | FR. ↓ | AU. ↑ | FR. ↓ | AU. ↑ | FR. ↓ | AU. ↑ |
| SSD+ [1] | 31.2 | 94.2 | **77.7** | **79.9** | 79.4 | 85.2 | 80.9 | 84.1 | 66.6 | 86.2 | 67.2 | 85.9 |
| CSI [2] | 44.5 | 92.7 | 79.1 | 76.3 | 75.6 | 83.8 | 76.6 | **85.0** | 61.6 | 86.5 | 67.5 | 84.8 |
| CIDER [3] | 23.1 | 95.2 | 79.6 | 73.4 | 16.2 | 96.3 | 71.7 | 83.0 | 43.9 | 90.4 | 46.9 | 87.7 |
| CIDER+VRA | **20.5** | **95.5** | 78.9 | 71.0 | **11.8** | **97.4** | **61.3** | 83.5 | **36.4** | **92.1** | **41.9** | **87.9** |

## B Compatibility with VIT[49]

Previous works [8, 10, 27] usually only consider the condition where the activation values are non-negative. To further illustrate the potential of VRA, we conduct experiments on VIT [49] with a large number of negative activation values.

For positive values, ReAct tries to increase $\mathbb{E}_{in}[g(z)] - \mathbb{E}_{out}[g(z)]$. It uses an operation to make OOD data suppress more than ID data. Therefore, for negative values, we should also make OOD data suppress more than ID data. This process results in an increase of $\mathbb{E}_{in}[-g(z)] - \mathbb{E}_{out}[-g(z)]$. To unify the positive and negative cases, we should maximize $\mathrm{sgn}(z)(\mathbb{E}_{in}[g(z)] - \mathbb{E}_{out}[g(z)])$, where $\mathrm{sgn}(\cdot)$ is the sign function. Similar to Eq. 3∼13, we can get the optimal activation:

$$g^*(z) = z + \lambda \mathrm{sgn}(z) \left( 1 - \frac{p_{out}(z)}{p_{in}(z)} \right).$$

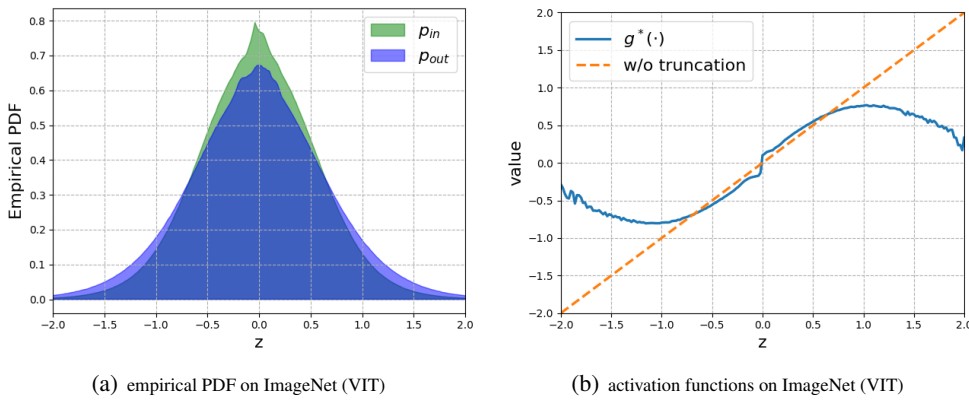

(a) empirical PDF on ImageNet (VIT)       (b) activation functions on ImageNet (VIT)

Figure 3: Empirical PDFs for $p_{in}(\cdot)$ and $p_{out}(\cdot)$ and visualization of different activation functions. We use VIT for ImageNet.

We visualize the optimal function $g^*(\cdot)$ on different ID and OOD data in Figure 3. We observe a different $g^*(\cdot)$ in VIT compared to ReLU-based backbones. Specifically, we should amplify activations with low absolute features and suppress activations with high absolute features. To mimic this operation, we design a new piecewise function called VRA-VIT:

$$\text{VRA-VIT}(z) = \begin{cases} -\alpha, z \leq -\alpha \\ \beta z, -\alpha \leq z \leq \alpha \\ \alpha, z \geq \alpha \end{cases},$$

Table 10: **Compatibility with VIT.** We conduct experiments on ImageNet.

| Backbone | Method | iNaturalist | | SUN | | Places | | Textures | | Average | |
|---|---|---|---|---|---|---|---|---|---|---|---|
| | | FR. ↓ | AU. ↑ | FR. ↓ | AU. ↑ | FR. ↓ | AU. ↑ | FR. ↓ | AU. ↑ | FR. ↓ | AU. ↑ |
| VIT-B-16 | Energy | 64.08 | 79.24 | 72.77 | 70.24 | 74.31 | 68.44 | 58.47 | 79.30 | 67.41 | 74.30 |
| | Energy + ReAct | 61.52 | 85.66 | 70.87 | 77.40 | 72.16 | 75.84 | 55.41 | 84.05 | 64.99 | 80.74 |
| | Energy + VRA-VIT | **55.80** | **89.96** | **66.42** | **84.02** | **68.29** | **82.46** | **53.44** | **86.57** | **60.99** | **85.76** |
| VIT-B-32 | Energy | 74.16 | 81.73 | 81.72 | 72.58 | 81.77 | 71.47 | 69.50 | 79.81 | 76.69 | 76.40 |
| | Energy + ReAct | 73.39 | 84.23 | 80.08 | 76.01 | 80.63 | 74.69 | 66.77 | 82.73 | 75.22 | 79.41 |
| | Energy + VRA-VIT | **65.49** | **88.03** | **76.88** | **80.63** | **76.49** | **79.59** | **65.98** | **84.20** | **71.21** | **83.11** |
| VIT-L-16 | Energy | 60.16 | 80.31 | 75.23 | 69.77 | 78.21 | 69.02 | 60.30 | 79.25 | 68.48 | 74.59 |
| | Energy + ReAct | **55.15** | 87.56 | 77.21 | 77.65 | 73.98 | 77.07 | **58.71** | 84.37 | **64.99** | 81.67 |
| | Energy + VRA-VIT | 57.24 | **90.49** | **71.07** | **82.84** | **71.44** | **82.31** | 62.45 | **86.16** | 65.55 | **85.45** |
| VIT-L-32 | Energy | 69.15 | 78.43 | 76.32 | 70.71 | 77.86 | 69.39 | 65.53 | 77.41 | 72.22 | 73.98 |
| | Energy + ReAct | 68.54 | 83.12 | 74.58 | 76.51 | 76.32 | 75.09 | 63.00 | 83.12 | 70.61 | 79.09 |
| | Energy + VRA-VIT | **61.00** | **89.61** | **71.21** | **83.14** | **71.65** | **82.02** | **62.70** | **84.82** | **66.64** | **84.75** |

where $\alpha > 0$ controls the threshold for determining low and high activations, and $\beta > 0$ controls the gradient. Results in Table 10 show the effectiveness of VRA, where we set $\alpha = 1$ and $\beta = 1.5$.

# C    More variance of VRA

The theoretical derivation of VRA and analysis of ID and OOD activations allow us to design other variants of VRA. In this section, we denote these new variants as VRA-B ( *binary* ) and VRA-G (change the *gradient* of the medium):

$$\text{VRA-B}(z) = \begin{cases} 0, z \leq \alpha \\ \alpha, z > \alpha \end{cases},$$

$$\text{VRA-G}(z) = \begin{cases} 0, z < \alpha \\ kz, \alpha \leq z \leq \beta \\ \beta, z > \beta \end{cases},$$

where $\alpha$ and $\beta$ are thresholds and $k$ controls the scaling factor. For VRA-G, we treat the $\eta_\alpha$-quantile (or $\eta_\beta$-quantile) of activations estimated on ID data as $\alpha$ (or $\beta$), in line with VRA. In Table 11, we investigate the performance of different VRA-based variants. For a fair comparison, we use the same ID data, OOD data, and backbone. Experimental results demonstrate that VRA++ outperforms other variants in OOD detection.

Table 11: **Performance of different VRA-based variants.** We use DenseNet-101 for CIFRA-10.

| Method | CIFAR-10 | |
|---|---|---|
| | FPR95 ↓ | AUROC ↑ |
| VRA-B ($\alpha = 0.5$) | 30.01 | 94.02 |
| VRA-B ($\alpha = 0.6$) | **20.37** | **95.84** |
| VRA-B ($\alpha = 0.7$) | 21.18 | 95.64 |
| VRA-B ($\alpha = 0.8$) | 35.84 | 93.31 |
| VRA-G ($k = 0.5$) | 28.14 | 94.12 |
| VRA-G ($k = 0.8$) | 19.62 | 96.11 |
| VRA-G ($k = 1.0$) | 17.74 | 96.47 |
| VRA-G ($k = 1.2$) | 17.32 | 96.64 |
| VRA-G ($k = 1.5$) | **16.81** | **96.66** |
| VRA-G ($k = 2.0$) | 17.05 | 96.56 |

# D    Discussion with BATS[50] and RankFeat[51]

BATS[50] is an OOD detection method that relies on BatchNorm. The main difference between BATS and our VRA lies in the following aspects. (1) BATS exploits the phenomenon that the features of the

training data more frequently fall in the interval $[\mu - \lambda\sigma, \mu + \lambda\sigma]$. Differently, VRA does not depend on BatchNorm. It tries to find the theoretically optimal operation to maximize the gap between ID and OOD. (2) BATS relies on BatchNorm and uses the mean and standard deviation in BatchNorm to determine thresholds for low and high activations. Differently, VRA does not rely on BatchNorm and determines thresholds for low and high activations using quantiles $\eta_\alpha$ and $\eta_\beta$. Experimental results in Table 6 show that VRA can achieve good performance no matter with or without BatchNorm (see VGG-16 and VGG-16-BN). Therefore, our VRA has better compatibility with different backbones compared with BATS. (3) BATS only suppresses high and low activations, but VRA+ and VRA++ further amplify intermediate activations. According to Table 12, this amplification process further improves the OOD detection performance.

Table 12: **Comparison with BATS**. For ImageNet, we use the pretrained ResNet50. For CIFAR, we use the pre-trained DenseNet-121.

| Method | CIFAR-10 | | CIFAR-100 | | ImageNet | |
|---|---|---|---|---|---|---|
| | FPR95 $\downarrow$ | AUROC $\uparrow$ | FPR95 $\downarrow$ | AUROC $\uparrow$ | FPR95 $\downarrow$ | AUROC $\uparrow$ |
| BATS | 24.30 | 95.32 | 59.32 | 86.79 | 27.11 | 94.28 |
| VRA | 17.74 | 96.47 | 47.12 | 90.21 | 25.49 | 94.57 |
| VRA+ | 15.89 | **96.90** | 43.31 | 90.61 | 23.32 | 94.96 |
| VRA++ | **15.52** | 96.87 | **35.20** | **91.80** | **18.63** | **95.75** |

RankFeat [51] finds that the feature maps of OOD samples often have large singular values, and removing these values are beneficial to OOD detection. We compare VRA with RankFeat, and the experimental results are shown in Table 13. We observe that our method has a significant improvement compared to RankFeat, especially on the OOD datasets iNaturalist and Texture.

Table 13: **Comparison with RankFeat**. We use ResNetv2-101 for ImageNet. The result of RankFeat is reported from RankFeat[51].

| Method | iNaturalist | | SUN | | Places | | Textures | | Average | |
|---|---|---|---|---|---|---|---|---|---|---|
| | FR. $\downarrow$ | AU. $\uparrow$ | FR. $\downarrow$ | AU. $\uparrow$ | FR. $\downarrow$ | AU. $\uparrow$ | FR. $\downarrow$ | AU. $\uparrow$ | FR. $\downarrow$ | AU. $\uparrow$ |
| RankFeat (Block 4) | 46.54 | 81.49 | **27.88** | 92.18 | **38.26** | 88.34 | 46.06 | 89.33 | 39.69 | 87.84 |
| RankFeat (Block 3) | 49.61 | 91.42 | 39.91 | 92.01 | 51.82 | 88.32 | 41.84 | 91.44 | 45.80 | 90.80 |
| RankFeat (Block 3+4) | 41.31 | 91.91 | 29.27 | **94.07** | 39.34 | **90.93** | 37.29 | 91.70 | 36.80 | 92.15 |
| VRA | 27.26 | 95.68 | 34.53 | 93.27 | 47.31 | 90.19 | 30.69 | 94.22 | 34.95 | 93.34 |
| VRA+ | **20.81** | **97.70** | 32.89 | 92.68 | 45.83 | 90.01 | **23.88** | **95.43** | **30.85** | **93.71** |

# E  Upper Bound Analysis

Similar to RankFeat, we provide theoretical analysis from the perspective of boundary analysis. Consider output logits $f(\mathbf{z}) = \mathbf{W}\mathbf{z} + \mathbf{b}$, where $\mathbf{z}$ is the feature vector of the penultimate layer. We denote the $i^{th}$ element in $f(\mathbf{z})$ as $f_i$, and $c$ is the number of categories. We compute the energy score of the output logits $f(\mathbf{z})$ as the OOD score, consistent with RankFeat:

$$S(\mathbf{z}) = \log \sum_{i=1}^{c} e^{f_i} = \log \sum_{i=1}^{c} e^{f_i - \max_{i=1}^{c} f_i} + \max_{i=1}^{c} f_i \leq \log c + \|f(\mathbf{z})\|_\infty,$$

For any norm $\|\cdot\|_p (p > 0)$, there exist $k_1 > 0$ and $k_2 > 0$ satisfying:

$$k_1 \|f(\mathbf{z})\|_p \leq \|f(\mathbf{z})\|_\infty \leq k_2 \|f(\mathbf{z})\|_p.$$

Then, we get:

$$S(\mathbf{z}) \leq k_2 \|\mathbf{W}\mathbf{z} + \mathbf{b}\|_p + \log c.$$

According to the triangle inequality $\|\mathbf{W}\mathbf{z} + \mathbf{b}\|_p \leq \|\mathbf{W}\mathbf{z}\|_p + \|\mathbf{b}\|_p$ and consistence of matrix norms $\|\mathbf{W}\mathbf{z}\|_p \leq \|\mathbf{W}\|_p \|\mathbf{z}\|_p$, we can get:

$$S(\mathbf{z}) \leq k_2 \|\mathbf{W}\|_p \|\mathbf{z}\|_p + k_2 \|\mathbf{b}\|_p + \log c.$$

According to the above inequality, maximizing $\mathbb{E}_{\text{in}}[z] - \mathbb{E}_{\text{out}}[z]$ is equivalent to maximizing the upper bound gap between ID and OOD. Therefore, the basic idea of VRA is similar to RankFeat. Specifically, RankFeat removes the rank-1 matrix from the high-level feature and increases the upper bound gap between ID and OOD; VRA maximizes $\mathbb{E}_{\text{in}}[z] - \mathbb{E}_{\text{out}}[z]$ to maximize the upper bound gap between ID and OOD. Therefore, we can prove VRA from the similar perspective of RankFeat.

# F   Visualization on CIFAR

Similar to Figure 1, we plot the activation distribution of ID and OOD data for the DenseNet-101 trained on CIFAR. As shown in Figure 4, we observe a similar distribution pattern to the ResNet-50 trained on ImageNet. Our designed modification function also exhibits the property of suppressing both ends and amplifying in the middle.

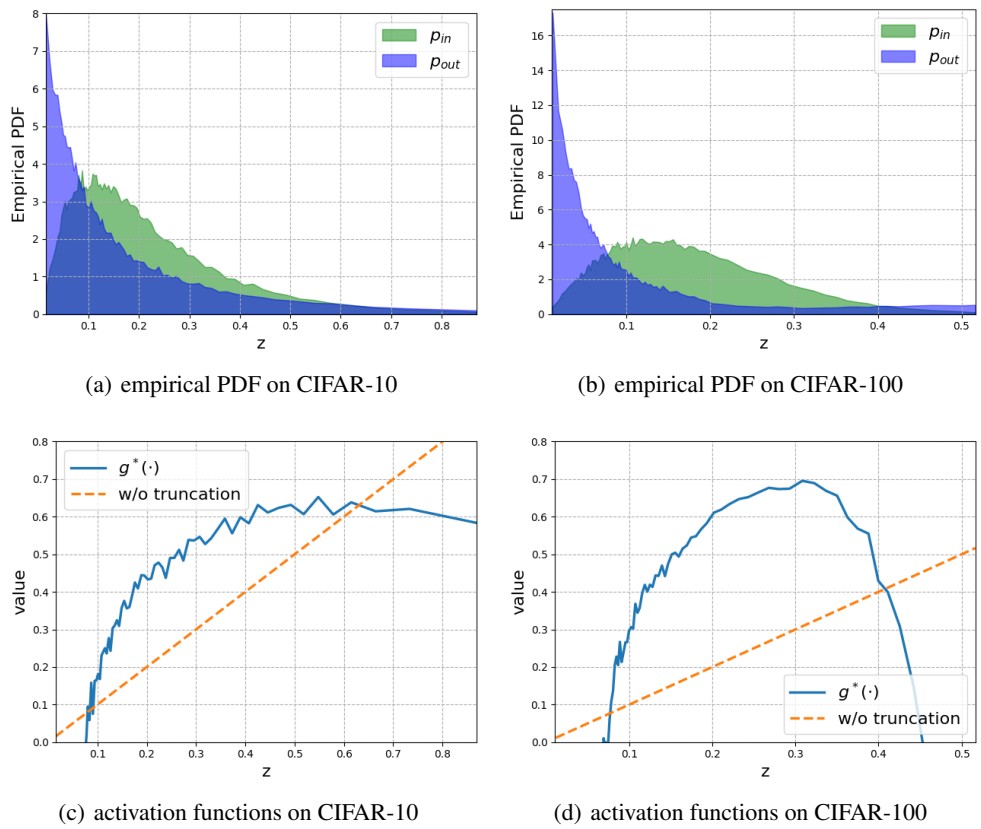

(a) empirical PDF on CIFAR-10   (b) empirical PDF on CIFAR-100

(c) activation functions on CIFAR-10   (d) activation functions on CIFAR-100

Figure 4: Empirical PDFs for $p_{\text{in}}(\cdot)$ and $p_{\text{out}}(\cdot)$ and visualization of different activation functions. We treat CIFAR as ID data and use DenseNet-101 as the backbone.

# G   Distribution of Energy Score

We plot the distribution of energy scores before and after VRA modification in Figure 5. After using VRA modification, the overlap between the energy scores of ID and OOD data becomes less. This phenomenon indicates that VRA makes it easier to distinguish ID and OOD data.

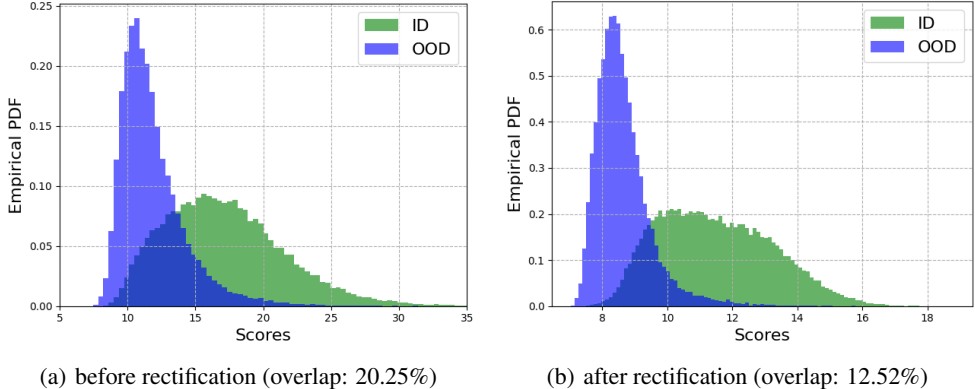

(a) before rectification (overlap: 20.25%)    (b) after rectification (overlap: 12.52%)

Figure 5: Distribution of scores before and after variational rectification. We treat ImageNet as ID data and use ResNet-50 as the backbone.

