# OpenReview forum: "VRA: Variational Rectified Activation for Out-of-distribution Detection"
_NeurIPS.cc/2023/Conference — NeurIPS 2023 poster_

### Official Review · Reviewer_JjUp · 2023-06-17

**Soundness:** 3 good
**Presentation:** 3 good
**Contribution:** 3 good
**Rating:** 6
**Confidence:** 5

**Summary:**

This paper conducts a fine-grained analysis of ReAct, a simple but effective OOD method by rectifying high activation. The authors perform a variational method to analyze the optimal activation function by clipping and amplifying features. Strong empirical results are presented to support the effectiveness of the proposed methods.

**Strengths:**

1. The proposed method is well-motivated both intuitively and theoretically. ReAct presented a promising solution to distinguish OOD samples by clipping activation. So it is natural to perform a fine-grained analysis of how to better clip activations. Actually, ASH [1] has a similar idea but this paper takes a step further to find the optimal solution of clipping from a theoretical point of view. The variational method seems valid and the assumption of Hilbert space is mostly valid for activations of deep neural networks. The theoretical analysis well motivates the methodology and shaping.

2. The authors propose two different variants of the method: one by simply clipping as React, and the other with middle activation amplification. This allows more possibilities to perform OOD detection.

3. The empirical results are very strong. In the ImageNet benchmark, it even outperforms MOS which is a training-needed approach.


>[1] Extremely Simple Activation Shaping for Out-of-Distribution Detection. ICLR 23.

**Weaknesses:**

1. My main concern is that the methodology seems to need to calculate $\rho_{out}$ in advance. However, the statistics of OOD data are usually unknown to the model. So having access to the density might reveal some information about OOD data, which might be unfair to the baselines. This might limit the real-world usage of the method.

2. There are some very relevant OOD papers missing in the reference [1,2,3,4]. As stated before, ASH[1] actually has done the similar fine-grained analysis of ReAct. Furthermore, the concept of shaping in ASH covers more operations such as pruning, binary, and scale. RankFeat [2] is also motivated by ReAct and removes the dominant single value of the last feature map. Moreover, it analyzes the lower bound of their method and ReAct. The bound analysis seems to apply to the method of this paper. The comparison and discussion with [1,2] are thus needed in the paper.

3. Some more operations such as binary and scaling can be explored. Currently, only amplification and pruning are supported.

>[1] Extremely Simple Activation Shaping for Out-of-Distribution Detection. ICLR 23.
>
>[2] RankFeat: Rank-1 Feature Removal for Out-of-distribution Detection

**Questions:**

Please see weakness and limitations.

**Limitations:**

My primary concern is the availability of OOD statistics required by this method. I suggest the authors at least explicitly mention it in the limitation section.

---

> ### Author Rebuttal · Authors · 2023-08-08
>
> # Response to Reviewer JjUp
> We thank the reviewer's appreciation of the writing and sufficient experimental results. We try to address your concerns as follows:
>
> **Q1**: My main concern is that the methodology seems to need to calculate $p_{out}$ in advance. However, the statistics of OOD data are usually unknown to the model. So having access to the density might reveal some information about OOD data, which might be unfair to the baselines. My primary concern is the availability of OOD statistics required by this method. I suggest the authors at least explicitly mention it in the limitation section.
>
> **A1**: Thanks for your valuable comments. As you pointed out, VRA does need to know $p_{out}$. If we know $p_{out}$ in advance, we can achieve near-perfect OOD detection results (see Table 4). But in real-world scenarios, we cannot obtain real OOD data in advance. For a fair comparison with baselines, we use Gaussian noise images as virtual OOD data in our implementation (see Section 3.1). According to Tables 1$\sim$2, we can still achieve state-of-the-art performance in OOD detection. Therefore, our comparison with baselines is fair. In the revised paper, we will add more discussions in the limitation section.
>
> **Q2**: There are some very relevant OOD papers missing in the reference. As stated before, ASH actually has done a similar fine-grained analysis of ReAct. Furthermore, the concept of shaping in ASH covers more operations such as pruning, binary, and scale. RankFeat is also motivated by ReAct and removes the dominant single value of the last feature map. The comparison and discussion with are thus needed in the paper.
>
> **A2**: Thank you for your valuable comments. Although VRA, ASH and RankFeat share some similarities, they are quite different in motivation, method, and results:
>
> *Motivation*: ASH argues that representations produced by over-parameterized neural networks are excessive for the task at hand, and therefore could be greatly simplified without much deterioration on the original performance while resulting
> in a surprising gain in OOD detection. RankFeat observes that the OOD feature matrix tends to have a larger dominant singular value than the ID feature, and the class predictions of OOD samples are largely determined by it. Different from ASH and RankFeat, VRA tries to find the theoretically optimal operation to maximize the gap between ID and OOD. Therefore, the motivations behind these methods are quite different.
>
> *Method*: Although all methods are extended from ReAct, their implementation process is different. ASH uses sample-wise pruning to remove a majority of the activation of the entire representation. Same to ASH, RankFeat is a also sample-wise pruning strategy. It removes the rank-1 matrix composed of the largest singular value and the associated singular vectors from the high-level feature. Differently, VRA follows ReAct and uses feature-wise pruning. According to Tables 1$\sim$2, our method can achieve the best results among all baselines.
>
> *Results*: We compare the performance of VRA-based methods with ASH and RankFeat. For a fair comparison, we use the same ID data, OOD data, and network architecture. Experimental results in Tables 1$\sim$2 demonstrate that VRA-based methods outperform previous works in OOD detection, verifying the effectiveness of our method. We will add these comparison results in the revised paper.
>
> **Table 1: Comparison with ASH. We use DenseNet-101 for CIFAR and ResNet-50 for ImageNet.**
>
> |Method|CIFAR-10(FR/AU)|CIFAR-100(FR/AU)|ImageNet(FR/AU)|Average(FR/AU)|
> |-|-|-|-|-|
> |ASH-P|23.45/95.22|64.53/82.71|50.32/89.04|46.10/88.99|
> |ASH-B|20.23/96.02|48.73/88.04|22.73/95.06|30.56/93.04|
> |ASH-S|15.05/96.61|41.40/90.02|22.80/95.12|26.42/93.92|
> |VRA|17.74/96.47|47.12/90.21|25.49/94.57|30.12/93.08|
> |VRA+|15.89/96.90|43.31/90.61|23.32/94.96|27.51/94.16|
> |VRA++|15.52/96.87|35.20/91.80|18.63/95.75|23.12/94.81|
>
> **Table 2: Comparison with RankFeat. We use ResNetv2-101 for ImageNet.**
>
> | Method   | FR/ AU |
> | - | - |
> | RankFeat (Block 4)|39.69/87.84|
> | RankFeat (Block 3)|45.80/90.80|
> | RankFeat (Block 3+4)|36.80/92.15|
> | VRA|34.95/93.34|
> | VRA+|**30.85**/**93.71**|
>
> **Q3**: RankFeat analyzes the lower bound of their method and ReAct. The bound analysis seems to apply to the method of this paper.
>
> **A3**: Thanks for your valuable comments. We have followed a similar method for analysis, but the space here is too small to write, and we will show it in our subsequent discussion and appendix.
>
> **Q4**: Some more operations such as binary and scaling can be explored. Currently, only amplification and pruning are supported.
>
> **A4**: Good idea! In this paper, we have not considered more operations such as binary and scaling. We denote these new variants as VRA-B and VRA-G:
> \begin{equation}
>     \text{VRA-B}(z)=\begin{cases}
>         0, z \leq \alpha \nonumber  \\\\
>         \alpha, z> \alpha\nonumber
>     \end{cases},
> \end{equation}
>
> \begin{equation}
>     \text{VRA-G}(z)=\begin{cases}
>         0, z< \alpha\nonumber \\\\
>         kz, \alpha \leq z \leq \beta \nonumber \\\\
>         \beta, z > \beta \\\\
>     \end{cases},
> \end{equation}
> where $\alpha$ and $\beta$ are thresholds and $k$ controls the scaling factor. For VRA-G, we treat the $\eta_\alpha$-quantile (or $\eta_\beta$-quantile) of activations estimated on ID data as $\alpha$ (or $\beta$), in line with VRA. In Table 3, we investigate the performance of different VRA-based variants. Experimental results demonstrate that VRA++ outperforms other variants in OOD detection.
>
> **Table 3: Performance of different VRA-based variants with DenseNet-101 on CIFAR-10.**
> |Method|FR/AU|
> |-|-|
> |VRA|17.74/96.47|
> |VRA+|15.89/**96.90**|
> |VRA++|**15.52**/96.87|
> |VRA-B$(\alpha=0.5)$|30.01/94.02|
> |$\alpha=0.6$|**20.37**/**95.84**|
> |$\alpha=0.7$|21.18/95.64|
> |$\alpha=0.8$|35.84/93.31|
> |VRA-G(k=0.5)|28.14/94.12|
> |k=0.8|19.62/96.11|
> |k=1.0|17.74/96.47|
> |k=1.2|17.32/96.64|
> |k=1.5|**16.81**/**96.66**|
> |k=2.0|17.05/96.56|

---

> > ### Comment · Reviewer_JjUp · 2023-08-13
> > **Thanks for the response**
> >
> > Thanks for the response!
> >
> > The reply addresses most of my concerns, but I have one final question:
> >
> > >A3: Thanks for your valuable comments. We have followed a similar method for analysis, but the space here is too small to write, and we will show it in our subsequent discussion and appendix.
> >
> > Can the authors give the results of the analysis here? There is no need to write the full derivations but I am just interested in the final results of the analysis.

---

> > > ### Author Response · Authors · 2023-08-13
> > > **Thanks for the response！**
> > >
> > > Thank you for taking your precious time to read our rebuttal and give us a response.  Based on your valuable suggestions, we provide theoretical proof from the perspective of boundary analysis to answer your final question.
> > >
> > > Consider output logits $f(\mathbf{z})=\mathbf{W}\mathbf{z}+\mathbf{b}$, where $\mathbf{z}$ is the feature vector of the penultimate layer. We denote the $i^{th}$ element in $f(\mathbf{z})$ as $f_i$, and $c$ is the number of categories. We compute the energy score of the output logits $f(\mathbf{z})$ as the OOD score, consistent with RankFeat:
> > > \begin{align}
> > >     S(\mathbf{z}) = \log \sum_{i=1}^c e^{f_i} = \log \sum_{i=1}^c e^{f_i - \max_{i=1}^c{f_i}} + \max_{i=1}^c f_i \le \log c + \lVert f(\mathbf{z}) \rVert_{\infty}, \nonumber
> > > \end{align}
> > >
> > > For any norm $\lVert \cdot \rVert_{p} (p > 0)$, there exist $k_1>0$ and $k_2>0$ satisfying:
> > > \begin{align}
> > >     k_1 \lVert f(\mathbf{z}) \rVert_p \le \lVert f(\mathbf{z}) \rVert_{\infty} \le k_2 \lVert f(\mathbf{z}) \rVert_p.
> > > \end{align}
> > >
> > > Then, we get:
> > > \begin{align}
> > >     S(\mathbf{z}) \le k_2 \lVert \mathbf{W}\mathbf{z}+\mathbf{b} \rVert_{p} + \log c.  \nonumber
> > > \end{align}
> > >
> > > According to the triangle inequality $\lVert \mathbf{W}\mathbf{z}+\mathbf{b} \rVert _p \leq  \lVert \mathbf{W}\mathbf{z}\rVert _p + \lVert \mathbf{b} \rVert_p $ and consistence of matrix norms $\lVert \mathbf{W}\mathbf{z}\rVert _p \leq \lVert \mathbf{W}\rVert _p \lVert \mathbf{z} \rVert_p$, and set $p = 1$, we can get:
> > >
> > > \begin{align}
> > >     S(\mathbf{z}) \le k_2  \lVert  \mathbf{W} \rVert _1  \lVert \mathbf{z}  \rVert _1 +  k_2  \lVert  \mathbf{b}  \rVert_1 + \log c.
> > > \end{align}
> > >
> > > According to the above inequality, maximizing $E_{in}[z] - E_{out}[z]$ in VRA means  increasing the upper bound gap  between ID and OOD.
> > >
> > > For a clear comparison we write the results in RankFeat:
> > > \begin{align}
> > >     S(\mathbf{z}) \le K\lVert \mathbf{W}\rVert_{\infty} (\sum_{i=1}^N s_i) + \lVert \mathbf{b} \rVert_{\infty} + \log c,
> > > \end{align}
> > > where $K>0$ and $s_i$ is the singular value of high-level feature map. Specifically, RankFeat "removes the rank-1 matrix from the high-level feature", which means replace $\sum_{i=1}^Ns_i$ with $\sum_{i=1}^N s_i - s_1$, where $s_1$ is the largest singular value. Then RankFeat think "OOD feature usually has a much larger $s_1$", so the operation "removes
> > > the rank-1 matrix" in RankFeat can increases the  upper bound  gap  between ID and OOD; VRA maximizes $E_{in}[z] - E_{out}[z]$ to also increase the  upper bound  gap between ID and OOD. But the upper bound is different between VRA and RankFeat.
> > >
> > > Above is the analysis of VRA from the similar perspective of RankFeat.

---

> > > > ### Comment · Reviewer_JjUp · 2023-08-13
> > > >
> > > > Thanks for the response!
> > > >
> > > > This analysis is very interesting, and further unifies previous works (RankFeat and ReAct).
> > > >
> > > > My concerns have been addressed. I would increase the score by one level.

---

> > > > > ### Author Response · Authors · 2023-08-14
> > > > > **Thank you !**
> > > > >
> > > > > Thank you very much for your valuable review !

---

### Official Review · Reviewer_J9PQ · 2023-07-06

**Soundness:** 3 good
**Presentation:** 2 fair
**Contribution:** 2 fair
**Rating:** 4
**Confidence:** 5

**Summary:**

The authors propose a technique called "Variational Rectified Activation (VRA)", which simulates these suppression and amplification operations using piecewise functions. Theoretical analysis is provided to illustrated VRA. Extensive experiments demonstrate the effectiveness and generalization for the proposed method.

**Strengths:**

1. It is technically sound to use variational methods to find the optimal function that maximizes the gap between ID and OOD.
2. The whole method is simple and effective, and performs well in different datasets and models. The proposal is also compatible with different scoring functions.
3. The paper is well written, well-structured and easy to understand.


**Weaknesses:**

1. The proposed method of simultaneously truncating high and low activations is very similar to the existing method in [1], which also corrects features by suppressing high and low activations, thus limiting the novelty of this proposal.
2. The formula in Eq.3 confuses me a bit. Why maximizing Ein(z) - Eout(z) maximizes the gap between ID and OOD. Why not maximize the squared difference? What happens if you maximize Eout(z) - Ein(z)?
3. The VRA piecewise function is designed based on the feature histograms in one model and one ID dataset, which may not hold for other datasets and model architectures. For example，for a Transformer-based model, the features in the penultimate layer contains many negative values，how the piecewise function should be in Transformer-based function.

&nbsp;

[1] Boosting Out-of-distribution Detection with Typical Features. NeurIPS 2022.

**Questions:**

1. Please compare the VRA framework with [1] and discuss the similarities and differences between the two.
2. In Fig.1, I'm curious about how the optimal function is look like if you use ViT as the classification model, and CIFAR as ID dataset. Could you provide more visualizations in the appendix.

&nbsp;

[1] Boosting Out-of-distribution Detection with Typical Features. NeurIPS 2022.

**Limitations:**

Please see the comments above.

---

> ### Author Rebuttal · Authors · 2023-08-08
>
> # Response to Reviewer J9PQ
> We sincerely thank the reviewer for your detailed comments and suggestions. We try to address each comment as satisfactorily as possible:
>
> **Q1**: The proposed method of simultaneously truncating high and low activations is very similar to the existing method in BATS, which also corrects features by suppressing high and low activations, thus limiting the novelty of this proposal. Please compare the VRA framework with BATS and discuss the similarities and differences between the two.
>
> **A1**: Thank you for your valuable comments. Although VRA and BATS  share some similarities, they are quite different in motivation, method, and results:
>
> *Motivation*: BATS relies on BatchNorm. For OOD detection, BATS exploits the phenomenon that the features of the training data more frequently fall in the interval $[ \mu -\lambda  \sigma, \mu + \lambda \sigma ]$. Differently, VRA does not depend on BatchNorm. It tries to find the theoretically optimal operation to maximize the gap between ID and OOD.
>
> *Method*: BATS relies on BatchNorm and uses mean and std to determine thresholds for low and high activations. Differently, VRA does not rely on BatchNorm and determines thresholds for low and high activations by quantiles. And the supressing is different, we suppress towards zero, they suppress towards mean. Experimental results in Table 6 of our main text show that VRA can achieve good performance no matter with or without BatchNorm (see VGG-16 and VGG-16-BN). Therefore, our VRA has better compatibility with different backbones compared with BATS. In addition, VRA+ and VRA++ further amplify intermediate activations.  Table 1 shows amplification process further improves the OOD detection performance.
>
> *Results*: We further compare the performance of VRA-based methods and BATS, we use DenseNet-121 for CIFAR and ResNet50 for ImageNet. Experimental results in Table 1 demonstrate that VRA-based methods outperform BATS in OOD detection, verifying the effectiveness of our method. We will add these comparison results in the revised paper.
>
> **Table 1: Comparison with BATS. (FR / AU)**
> | Method |CIFAR-10 | CIFAR-100 | ImageNet |
> | - | - | - | - |
> |BATS | 24.30 /95.32|59.32/86.79|27.11/94.28|
> |VRA|17.74/96.47|47.12/90.21|25.49/94.57|
> |VRA+ |15.89 /96.90|43.31/90.61|23.32/94.96|
> |VRA++|15.52/96.87|35.20/91.80|18.63/95.75|
>
> **Q2**: The formula in Eq. 3 confuses me a bit. Why maximizing $E_{in}(z) - E_{out}(z)$ maximizes the gap between ID and OOD. Why not maximize the squared difference? What happens if you maximize $E_{out}(z) - E_{in}(z)$?
>
> **A2**: The objective function $max  E{in}g(z) - E_{out}g(z)$ is derived from ReAct, a well-known and effective OOD detection technique. ReAct proves that increasing $E_{in}g(z) - E_{out}g(z)$ leads to better OOD detection performance. In this paper, we extend the idea of ReAct and propose a new operation VRA to maximize $E_{in}g(z) - E_{out}g(z)$. Experimental results prove that we can achieve better performance than ReAct in OOD detection.
>
> **Q3**: The VRA piecewise function is designed based on the feature histograms in one model and one ID dataset, which may not hold for other datasets and model architectures.
>
> **A3**: Thanks for your valuable comments. In fact, we design VRA based on the results of multiple models and multiple ID datasets. Due to page limitations, we did not include all visualization results in this paper. In Figure 2 (see attached PDF in Global Response), we provide more visualization results and obverse the same phenomenon. We will add these visualization results in the appendix.
>
> **Q4**: For a Transformer-based model, how the piecewise function should be in the Transformer-based function. I'm curious about how the optimal function looks like if you use ViT as the classification model. Could you provide more visualizations in the appendix?
>
> **A4**: Good question! Our core function comes from ReAct, which relies on ReLU-based backbones where the penultimate layer does not contain negative values. For the case of many negative values, we conduct the following analogy analysis.
>
> For positive values, ReAct tries to increase $E_{in}g(z) - E_{out}g(z)$. It uses an operation to make OOD data suppress more than ID data. Therefore, for negative values, we should also make OOD data suppress more than ID data. This process results in an increase of $E_{in}[-g(z)] - E_{out}[-g(z)]$. To unify the positive and negative cases, we should maximize $\mbox{sgn}(z)(E_{in}g(z) - E_{out}g(z))$, where $\mbox{sgn}(\cdot)$ is the sign function. Similar to Eq. 3$\sim$13, we can get the optimal activation:
> \begin{align}
>     g^*(z) = z + \lambda \mbox{sgn}(z) \left(1 - \frac{p_{out}(z)}{p_{in}(z)}\right).
> \end{align}
>
> We visualize the optimal function $g^*$ on different ID and OOD data. Due to page limitations, we will put these visualization results in the appendix. We observe a different $g^*$ in ViT compared to ReLU-based backbones. Specifically, we should amplify activations with low absolute features and suppress activations with high absolute features. To mimic this operation, we design a new piecewise function called VRA-ViT:
> \begin{equation}
>     \text{VRA-ViT}(z)=\begin{cases}
>         -\alpha, z \leq -\alpha \nonumber  \\\\
>         \beta z, -\alpha <  z < \alpha\nonumber \\\\
>         \alpha,z \ge \alpha
>         \nonumber
>     \end{cases},
> \end{equation}
> where $\alpha>0$ controls the threshold for determining low and high activations, and $\beta>0$ controls the gradient. Experimental results in Table 2 show the effectiveness of VRA.
>
> **Table 2: Compatibility with ViT on ImageNet.**
> |Method|Backbone|FR / AU|
> | - | - | - |
> |Energy|B/16|67.41/74.30|
> |+ReAct||64.99?80.74|
> |+VRA-ViT||60.99/85.76|
> |Energy|B/32|76.69/76.40|
> |+ReAct||75.22/79.41|
> |+VRA-ViT||65.98/84.20|71.21/ 83.11|
> |Energy|L/16|68.48/74.59|
> |+ReAct||64.99 / 81.67|
> |+VRA-ViT||62.45 / 86.16| 65.55 / 85.45|
> |Energy|L/32|72.22 / 73.98 |
> |+ReAct||70.61/79.09 |
> |+VRA-ViT|| 66.64/ 84.75 |

---

> > ### Comment · Reviewer_J9PQ · 2023-08-19
> > **Response to Rebuttal**
> >
> > I would like to thank the authors for their rebuttal. But my main concerns have not been addressed.
> >
> > 1. The VRA optimal operation is more like a simple variant of existing feature clipping methods, such as ReAct[1], BAST[2], ASH[3]. ReAct clips high activations, BAST clips both low activations and high activations and ASH-B clips low activations. Although the author emphasizes the difference from BAST, in fact VRA only add three hyper-parameters, while BAST utilized mean and variance to get the min and max thresholds.
> > 2. There are still some problems in the interpretation of formula 3. ReAct clips the high activations and proves that the average activation reduction of OOD is larger than the reduction of ID, which is reasonable. Eq.3 is lack of insight, the author may provide more explanations about why the objective function (Eq.3 ) benefits OOD performance.
> > 3. The optimal operation is based on feature histograms of ID set and OOD set. For different models and OOD datasets, the estimated optimal operation functions might be different, which makes it difficult to obtain the optimal function. For example, the activation functions on iNaturalist, SUN and CIFAR-100 are quite different. Besides, it seems that the optimal function is estimated based on the test ID/OOD set, rather than a validation set.
> >
> > In general，It is interesting and novel to utilize variational method to estimate the optimal activation function. But there are some significant shortcomings, making it hard to be accepted by NeurIPS. I believe the paper can be significantly improved if the author could provide a more reasonable and insightful objective function, and estimate a robust operation function.
> >
> > - [1] ReAct: Out-of-distribution Detection with Rectified Activations.
> > - [2] Boosting Out-of-distribution Detection with Typical Features
> > - [3] Extremely Simple Activation Shaping for Out-of-Distribution Detection.

---

> > > ### Author Response · Authors · 2023-08-19
> > > **Thanks for your response.**
> > >
> > > We greatly appreciate your reply and want to address your concern as much as possible.
> > >
> > > **Q1**: The VRA optimal operation is more like a simple variant of existing feature clipping methods, such as ReAct, BAST, ASH. ReAct clips high activations, BAST clips both low activations and high activations and ASH-B clips low activations. Although the author emphasizes the difference from BAST, in fact VRA only add three hyper-parameters, while BAST utilized mean and variance to get the min and max thresholds.
> > >
> > > **A1**: Thank you very much for your valuable comments. In addition to our responses to Reviewer J9PQ (see A1) and Reviewer JjUp (see A2), we also try to restate the difference between VRA-based methods with existing approaches. In this paper, we extend ReAct and propose new activation functions in OOD detection. VRA-based approaches are not simple variants of existing approaches but have strong motivations and theoretical guarantees. Compared with ReAct, BAST, and ASH which merely truncate low and high activations, we further prove the necessity of amplifying intermediate activations. Such a simple modification can achieve state-of-the-art OOD detection performance, which fully validates its effectiveness. As Reviewer JjUp comments, our VRA-based methods "allow more possibilities to perform OOD detection".
> > >
> > > **Q2**: There are still some problems in the interpretation of formula 3. ReAct clips the high activations and proves that the average activation reduction of OOD is larger than the reduction of ID, which is reasonable. Eq.3 is lack of insight, the author may provide more explanations about why the objective function (Eq.3 ) benefits OOD performance.
> > >
> > > **A2**: Thanks for your comments and we apologize for our unclear description. First, let's review the theoretical analysis in ReAct (see Section 5 of ReAct ). It proves that the rectification operation affects OOD activations more severely compared to ID activations and results in a large $ E_{out} ( z_i -  \bar{z_i} ) - E_{in}  ( z_i  -  \bar{z_i} ) $ (see Remark 1 of ReAct). The increased separation between OOD and ID activations can transfer to the output space as well (see Remark 2 of ReAct), thus enlarging the gap between OOD and ID score. Rather than just increasing separation like ReAct, this paper attempts to maximize the separation. Based on this motivation, we propose the objective function in Eq. 3. Experimental results on benchmark datasets also demonstrate the effectiveness of our method.
> > >
> > > **Q3**: The optimal operation is based on feature histograms of ID set and OOD set. For different models and OOD datasets, the estimated optimal operation functions might be different, which makes it difficult to obtain the optimal function. For example, the activation functions on iNaturalist, SUN and CIFAR-100 are quite different.
> > >
> > > **A3**: Thanks for your valuable comments. In Figure 2 (see attached PDF in Global Response) and Figure 1 (see the main text), we provide visualization results on multiple datasets and obverse the same phenomenon. From this phenomenon, we verify the necessity of suppressing abnormally low and high activations and amplifying intermediate activations. Although different models and OOD datasets might have different optimal operation functions, we can adjust the hyper-parameters in activation functions to approximate the optimal operation function.
> > >
> > > **Q4**: Besides, it seems that the optimal function is estimated based on the test ID/OOD set, rather than a validation set.
> > >
> > >
> > > **A4**: In fact, we use Gaussian noise images as the validation set for hyper-parameter tuning (see Implementation Details in the main text). If we choose the best hyper-parameter according to the test ID/OOD set, we can achieve even higher OOD detection performance. For example, we set $\eta_{\alpha} = 0.6$, $\eta_{\beta} = 0.95$ and report "17.74/96.47(FPR95/AUROC)'' in Table 1 of our main text. If we choose the best hyper-parameter based on the test ID/OOD set, we can get "16.78/96.63 (FPR95/AUROC)''.

---

### Official Review · Reviewer_Tdya · 2023-07-06

**Soundness:** 3 good
**Presentation:** 3 good
**Contribution:** 3 good
**Rating:** 5
**Confidence:** 4

**Summary:**

This work leverages the variational method to find optimal operation and proposes a new technique variational rectified activation (VRA) for out-of-distribution (OOD) detection. This paper finds the best operation for OOD detection and verifies the necessity of suppressing abnormally low and high activations and amplifying intermediate activations. The proposed VRA method is compatible with different network architectures and scoring functions. Extensive experiments on a number of benchmark datasets show the effectiveness of the proposed method.

**Strengths:**

1. This work tackles out-of-distribution (OOD) detection tasks, which is important for building reliable machine learning models in the real world.
2. The key idea of variational rectified activation that mimics suppression and amplification operations using piecewise functions is simple and flexible to compatible with different scoring functions and network architectures. This work also provides a theoretical understanding of the VRA method for OOD detection from the perspective of the variational method.
3. This paper conducts extensive experiments on several benchmark datasets to validate the effectiveness of the proposed VRA method.

**Weaknesses:**

1. Can the authors report the mean and std for the main results in the experiment section?
2. It would be interesting if the authors can compare the visualization of the distribution of ID and OOD uncertainty scores before and after variational rectification.

**Questions:**

Refer to the detailed comments on weaknesses.

**Limitations:**

This work does not present any potential negative societal impact.

---

> ### Author Rebuttal · Authors · 2023-08-08
>
> # Response to Reviewer Tdya
>
> We thank the reviewer's appreciation of our clear writing and sufficient experimental results. We try to address your concerns as follows:
>
> **Q1**: Can the authors report the mean and std for the main results in the experiment section?
>
> **A1**: Thanks for your valuable comments. Since the standard deviations are relatively small, we did not include them in the main results. In Table 1, we take the main results on CIFAR-10 as an example.
>
> **Table 1: Mean and std for the main results. We use DenseNet-101 for CIFAR-10.**
> | Method        | CIFAR-10 (FR $\downarrow$ / AU$\uparrow$) |
> | ------------- | ----------------------------------------- |
> | MSP           |  48.74$\pm$0.00 / 92.46$\pm$0.00     |
> | ODIN          |  24.57$\pm$0.00 / 93.71$\pm$0.00     |
> | Mahalanobis   |  31.42$\pm$0.00 / 89.15$\pm$0.00     |
> | Energy        |  26.55$\pm$0.00 / 94.57$\pm$0.00     |
> | ReAct         |  26.45$\pm$0.00 / 95.30$\pm$0.00     |
> | KNN           |  25.83$\pm$0.00 / 94.39$\pm$0.00     |
> | DICE          |  20.84$\pm$1.58 / 95.25$\pm$0.24     |
> | SHE           |  26.82$\pm$0.00 / 92.98$\pm$0.00     |
> | VRA  |  17.74$\pm$0.00 / 96.47$\pm$0.00     |
> | VRA+ |  15.85$\pm$0.00/ 96.91$\pm$0.00  |
>
>
>
>
>
> **Q2**: It would be interesting if the authors can compare the visualization of the distribution of ID and OOD uncertainty scores before and after variational rectification.
>
> **A2**: Thanks for your valuable suggestion, In Figure 1 (see attached PDF in Global Response), we visualize the distribution of ID and OOD uncertainty scores before and after variational rectification. We observe that our rectification operation can reduce the overlap between ID and OOD data, verifying the effectiveness of our strategy.

---

> > ### Comment · Reviewer_Tdya · 2023-08-21
> > **Response to Rebuttal**
> >
> > Thanks for the authors' rebuttal. I read the response and other reviewers' comments. After consideration, I decided to keep my original score.

---

> > > ### Author Response · Authors · 2023-08-22
> > > **Response to Reviewer Tdya**
> > >
> > > Thank you for your valuable comments. We decide to present our novelty, contribution, related work, and technology more clearly. We kindly ask the reviewer to reconsider the rating.
> > >
> > > Novelty and Contribution: To the best of our knowledge, this is the first work that exploits the variational approach to find the optimal operation and validates the necessity of amplifying intermediate activations in OOD detection. VRA-based approaches are not simple variants of existing works but have strong motivations and theoretical guarantees. As Reviewer JjUp comments, our VRA-based methods "allow more possibilities to perform OOD detection". Experimental results on multiple benchmark datasets demonstrate that our method outperforms existing post-hoc strategies. Therefore, this paper is novel and points out a new direction for OOD detection.
> > >
> > > Related Works: In this paper, we have taken the reviewers' suggestions and compared our VRA-based methods with more related works, including SSD+, CSI, CIDER, MaxLogit, HEAT, BATS, ASH, and RankFeat. Compared with existing works, our method is different in motivation and practical form (see our previous response). Such a simple strategy can achieve state-of-the-art OOD detection performance, which fully validates the effectiveness of our method. Please specify which relevant works we are missing. We can add more discussions and comparisons.
> > >
> > > Technology: We have taken the reviews' suggestions and proposed more variants for VRA, including VRA-G (see the response to Reviewer GN4S), VRA-ViT (see the response to Reviewer J9PQ), and VRA-B (see the response for Reviewer JjUp). Our method allows more possibilities for OOD detection.

---

### Official Review · Reviewer_GN4S · 2023-07-10

**Soundness:** 3 good
**Presentation:** 3 good
**Contribution:** 3 good
**Rating:** 4
**Confidence:** 3

**Summary:**

In this paper, the authors explore the out-of-distribution detection ability. They propose Variational Rectified Activation based on ReAct. Specifically, they suppress both the high and low values of the penultimate layer rather than only focusing on high values. Their performance on several benchmark dataset surpasses existing post-hoc strategies.

**Strengths:**

1. They give a theoretical analysis to the proposed method.
2. They compare VRA-based methods with competitive post-hoc strategies, and performs the best.

**Weaknesses:**

1. The theory part should be more clear about the relationship between the objective you minimize and the final score.
2. Please add more baselines to demonstrate the effectiveness of VRA, comparing your method with contrastive learning trained backbones (SSD+[1], CSI[2], CIDER[3]), and other state-of-the-art post-hoc methods (maxlogit[4] and HEAT[5]).



[1] https://openreview.net/forum?id=v5gjXpmR8J

[2] https://openreview.net/forum?id=o5RKoLQlK4olF

[3] https://openreview.net/forum?id=aEFaE0W5pAd

[4] https://arxiv.org/abs/1911.11132

[5] https://openreview.net/forum?id=tpCynHFviX

**Questions:**

1. In equation 3, why it is needed to maximally preserve the input? And with this term, why the optimal operation maximize the gap between ID and OOD?

2. In figure 1, your optimal g function does not seem to agree with VRA function. Have you ever try to adjust the gradient when z is between alpha and beta?

**Limitations:**

It seems that the paper did not include a limitation section or paragraph explicitly.

---

> ### Author Rebuttal · Authors · 2023-08-08
>
> # Response to Reviewer GN4S
> We thank the reviewer's appreciation of our theoretical analysis and rich experimental results. We try to address your concerns as follows:
>
> **Q1**: The theory part should be more clear about the relationship between the objective you minimize and the final score.
>
> **A1**: ReAct has proved that increasing the gap between ID and OOD (see Eq. 2) can improve the final score in OOD detection. We adopt the same objective function as ReAct and try to find the optimal operation to widen the gap. Therefore, the theoretical proof of the relationship between the objective function and final scores is equivalent to ReAct. Due to page limitations, we have not included these parts.
>
> **Q2**: Please add more baselines to demonstrate the effectiveness of VRA, comparing your method with contrastive learning trained backbones (SSD+, CSI, CIDER), and other state-of-the-art post-hoc methods (MaxLogit and HEAT).
>
> **A2**: Due to the good compatibility of VRA, for a fair comparison, we test the performance of VRA under the same experimental settings as the baselines (including ID data, OOD data, and network architecture). For contrastive-learning-based methods, we conduct experiments under the same settings as CIDER. For post-hoc methods, we conduct experiments under the same settings as HEAT. Experimental results in Tables 1$\sim$2 demonstrate that VRA outperforms existing OOD detection strategies.
>
>
> **Table1: Comparison with contrastive learning trained backbones. We use ResNet-34 for CIFAR-100.**
>
> | Method | FR $\downarrow$ / AU$\uparrow$ |
> | ------ | --------------------------------------- |
> | SSD+   | 67.2 / 85.9                             |
> | CSI    | 67.5 / 84.8                            |
> | CIDER  |  46.9 / 87.7                           |
> | VRA    | **41.9** / **87.9**                    |
>
>
>
> **Table 2: Comparison with post-hoc methods. We use ResNet-50 on ImageNet.**
>
> | Method   | FR $\downarrow$ / AU$\uparrow$ |
> | -------- | --------------------------------------- |
> | MaxLogit |  58.0 / 87.0                           |
> | HEAT     |  34.4 / 92.6                           |
> | VRA      | **25.5** / **94.6**                      |
>
>
> **Q3**: In equation 3, why it is needed to maximally preserve the input? And with this term, why the optimal operation maximize the gap between ID and OOD?
>
> **A3**: We apologize for this unclear description. In this paper, we treat $\max_g  \mathbb{E}{\text{in}}[g(z)] - \mathbb{E}{\text{out}}[g(z)]$ as the core objective function derived from ReAct and $\min_g  \mathbb{E}{\text{in}}[(g(z)-z)^2]$ as the regularization term. The motivation behind this regularization term is twofold: (1) Many post-hoc methods do not change the feature space but also achieve promising results on OOD detection (such as MSP, Energy, and ODIN). Therefore, $g(z)=z$ is an acceptable operation. (2) If we only maximize $\mathbb{E}{\text{in}}[g(z)] - \mathbb{E}{\text{out}}[g(z)]$ without the regularization term, the optimal solution will not exist. The proof is by contradiction. Suppose there is an optimal operation $g^*(\cdot)$. Then, we can easily get a better operation $2g^*(\cdot)$ with larger $\mathbb{E}{\text{in}}[g(z)] - \mathbb{E}{\text{out}}[g(z)]$. Therefore, this regularization term can guarantee the existence of the optimal solution. There may be better regularization terms. In the future, we will explore other regularization terms for OOD detection.
>
> In Eq. 14$\sim$15, we prove that the optimal operation $g^*(\cdot)$ driven by our objective function can widen the gap between ID and OOD by at least $\frac{1}{2\lambda} \mathbb{E}{\text{in}}[(g^*(z)-z)^2] \geq 0$. Therefore, adding this term does not change the core objective of VRA. Meanwhile, Table 4 (see the submitted paper) shows that $g^*(\cdot)$ can achieve near-perfect results on OOD detection. Therefore, although this regularization term slightly changes the objective function, it does not significantly affect the performance of OOD detection but ensures the existence of the optimal solution. Just like the regularization term in neural networks. Although the regularization term changes the loss function, it prevents overfitting and improves the generalization ability of neural networks.
>
> **Q4**: In figure 1, your optimal g function does not seem to agree with VRA function. Have you ever try to adjust the gradient when z is between alpha and beta?
>
> **A4**: Good idea! In this paper, VRA+ introduces a hyper-parameter $\gamma$ to amplify intermediate activations. However, we have not considered adjusting the gradient at $\alpha \leq z \leq \beta$. We denote this new variant as VRA-G:
> \begin{equation}
>     \text{VRA-G}(z)=\begin{cases}
>         0, z< \alpha\nonumber \\\\
>         kz, \alpha \leq z \leq \beta \nonumber \\\\
>         \beta, z > \beta \\\\
>     \end{cases},
> \end{equation}
> where $k$ controls the gradient. Same with VRA, we treat the $\eta_\alpha$-quantile (or $\eta_\beta$-quantile) of activations estimated on ID data as $\alpha$ (or $\beta$). It is worth noting that VRA-G with $k=1.0$ is equivalent to VRA. Experimental results in Table 3 show that setting the gradient a little steeper (i.e., $k>1$) can amplify intermediate activations and achieve better performance. But over-amplification leads to performance degradation.
>
>
>
> **Table 3: Performance of VRA-G. We use DenseNet-101 for CIFAR-10.**
>
> | Method          | CIFAR-10 (FR $\downarrow$ / AU$\uparrow$) |
> | --------------- | ----------------------------------------- |
> | VRA             | 17.74 / 96.47                        |
> | VRA+            | **15.89** / **96.90**                   |
> | VRA-G $(k=0.5)$ | 28.14 / 94.12                         |
> | VRA-G $(k=0.8)$ | 19.62 / 96.11                        |
> | VRA-G $(k=1.0)$ | 17.74 / 96.47                        |
> | VRA-G $(k=1.2)$ | 17.32 / 96.64                       |
> | VRA-G $(k=1.5)$ | **16.81** / **96.66**                 |
> | VRA-G $(k=2.0)$ | 17.05 / 96.56                        |

---

> > ### Comment · Reviewer_GN4S · 2023-08-21
> >
> > Thanks for the response. I appreciate the author(s)' effort in this work, but I also realize that the paper has some weaknesses, e.g., in the aspect of novelty, contribution, related work, and technology as other reviewers and I raised in the comments. So, I am inclined to maintain my initial rating.

---

> > > ### Author Response · Authors · 2023-08-21
> > > **Response to Reviewer GN4S**
> > >
> > > **Q1:** I appreciate the author(s)' effort in this work, but I also realize that the paper has some weaknesses, e.g., in the aspect of novelty, contribution, related work, and technology as other reviewers and I raised in the comments. So, I am inclined to maintain my initial rating.
> > >
> > > **A1:** Thank you for your valuable comments. To address your concerns, we decide to present our novelty, contribution, related work, and technology more clearly. We kindly ask the reviewer to reconsider the rating.
> > >
> > > *Novelty and Contribution:* To the best of our knowledge, this is the first work that exploits the variational approach to find the optimal operation and validates the necessity of amplifying intermediate activations in OOD detection. VRA-based approaches are not simple variants of existing works but have strong motivations and theoretical guarantees. As Reviewer JjUp comments, our VRA-based methods "allow more possibilities to perform OOD detection". Experimental results on multiple benchmark datasets demonstrate that our method outperforms existing post-hoc strategies. Therefore, this paper is novel and points out a new direction for OOD detection.
> > >
> > > *Related Works:* In this paper, we have taken the reviewers' suggestions and compared our VRA-based methods with more related works, including SSD+, CSI, CIDER, MaxLogit, HEAT, BATS, ASH, and RankFeat. Compared with existing works, our method is different in motivation and practical form (see our previous response). Such a simple strategy can achieve state-of-the-art OOD detection performance, which fully validates the effectiveness of our method. Please specify which relevant works we are missing. We can add more discussions and comparisons.
> > >
> > > *Technology:* We have taken the reviews' suggestions and proposed more variants for VRA, including VRA-G (see the response to Reviewer GN4S), VRA-ViT (see the response to Reviewer J9PQ), and VRA-B (see the response for Reviewer JjUp). Our method allows more possibilities for OOD detection.

---

### Author Rebuttal · Authors · 2023-08-08

# Global Response
Dear Reviewers, Area Chairs, and Program Chairs:

We would like to express our gratitude to all reviewers for taking their valuable time to review our paper. We sincerely appreciate all reviewers for their positive comments on our theoretical analysis and technique soundness. For example, Reviewer JjUp points out that "the theoretical analysis well motivates the methodology and shaping". Reviewer JjUp notes that the proposed method is "well-motivated both intuitively and theoretically". We also thank the reviewers for noting that this paper is "well-written, well-structured, and easy to understand" (J9PQ), the proposed method is "promising" (JjUp) and "allows more possibilities to perform OOD detection" (JjUp), the performance is "competitive" (GN4S), "effective" (Tdya, J9PQ) and "strong" (JjUp). Meanwhile, we appreciate the reviewers for pointing out the shortcomings. Your valuable comments help us improve this paper. We try to address each comment as satisfactorily as possible. In the response, we compare with more baselines (including SSD+, CSI, CIDER, MaxLogit, HEAT, BATS, ASH, and RankFeat), verify the compatibility of our method with more backbones (such as ViT), present more visualization results (including the distribution of ID and OOD uncertainty scores before and after variational rectification, and the optimal operation in CIFAR-10 and CIFAR-100), and study more VRA-based variants (such as VRA-B and VRA-G). Please find the responses to each reviewer’s comments below.

We kindly ask the reviewers to take the above clarifications into account when considering score adjustments. We welcome any further discussion with the reviewers.

Best regards,

Paper4986 Authors

---

### Decision · Program_Chairs · 2023-09-21

**Decision:**

Accept (poster)

**Comment:**

This paper has received relatively diverging reviews. Reviewers Tdya and JjUp consider that the paper has enough contributions to be accepted, while reviewers GN4S and J9PQ expressed certain concerns. In particular, reviewer GN4S initially asked for clarifications on several questions and additional baselines. After these questions were answered, reviewer GN4S further raised concerns in novelty, contribution, related work, and technology, but did not further detail these concerns nor follow up with more discussions. Reviewer J9PQ  mainly had concerns in novelty (difference between this work and ReAct/BAST/ASH) and want to get clarifications on Eq. 3 and the choice of hyper-parameters. The last two are sufficiently addressed by the author response. In addition, AC considers the novelty of this work enough upon reading through the paper and all the discussions. Even though the work is an extension of ReAct, the contributions are motivated and empirically backed by the experiments. The AC thus recommends acceptance of this work to NeurIPS.

The decision of this paper was communicated with the SAC, and was made based on the discussion.

The authors are highly encouraged to incorporate the feedback and rebuttal discussions/experiments into the final version and revise accordingly.